# Glycolipid-dependent and lectin-driven transcytosis in mouse enterocytes

Alena Ivashenka[1], Christian Wunder [1], Valerie Chambon[1], Roger Sandhoff [2], Richard Jennemann[2], Estelle Dransart[1], Katrina Podsypanina[3], Bérangère Lombard [4], Damarys Loew [4], Christophe Lamaze [5], Francoise Poirier[6], Hermann-Josef Gröne[7], Ludger Johannes [1✉] & Massiullah Shafaq-Zadah [1✉]

Glycoproteins and glycolipids at the plasma membrane contribute to a range of functions from growth factor signaling to cell adhesion and migration. Glycoconjugates undergo endocytic trafficking. According to the glycolipid-lectin (GL-Lect) hypothesis, the construction of tubular endocytic pits is driven in a glycosphingolipid-dependent manner by sugar-binding proteins of the galectin family. Here, we provide evidence for a function of the GL-Lect mechanism in transcytosis across enterocytes in the mouse intestine. We show that galectin-3 (Gal3) and its newly identified binding partner lactotransferrin are transported in a glycosphingolipid-dependent manner from the apical to the basolateral membrane. Transcytosis of lactotransferrin is perturbed in Gal3 knockout mice and can be rescued by exogenous Gal3. Inside enterocytes, Gal3 is localized to hallmark structures of the GL-Lect mechanism, termed clathrin-independent carriers. These data pioneer the existence of GL-Lect endocytosis in vivo and strongly suggest that polarized trafficking across the intestinal barrier relies on this mechanism.

---

[1] Institut Curie, Université PSL, U1143 INSERM, UMR3666 CNRS, Cellular and Chemical Biology Unit, Endocytic Trafficking and Intracellular Delivery Team, Paris, France. [2] Lipid Pathobiochemistry Group, German Cancer Research Center, Heidelberg, Germany. [3] Institut Curie, Université PSL, UMR144 CNRS, Cell Biology and Cancer, Paris, France. [4] Institut Curie, Université PSL, Mass Spectrometry and Proteomics Facility, Paris, France. [5] Institut Curie, Université PSL, U1143 INSERM, UMR3666 CNRS, Cellular and Chemical Biology Unit, Membrane Dynamics and Mechanics of Intracellular Signaling Team, Paris, France. [6] Institut Jacques Monod, UMR 7592 CNRS - Université Paris Diderot, 15 rue Hélène Brion, Paris, France. [7] Institute of Pharmacology, University of Marburg, Marburg, Germany. ✉email: ludger.johannes@curie.fr; massiullah.shafaq-zadah@curie.fr

Endocytic uptake of adhesion molecules, growth factor receptors, and other cargoes is controlled by numerous mechanisms, of which the clathrin pathway is the best characterized[1–3]. How cargoes are recruited, and the plasma membrane bent to form endocytic pits in the case of clathrin-independent endocytosis is currently a dynamic field of exploration[4,5].

Evidence has been provided that in some cases, tubular endocytic pits for clathrin-independent uptake into cells are induced by the interaction of pathogenic (e.g., Shiga and cholera toxins) or cellular lectins (e.g., galectins) with glycolipids[6–8]. The mechanism by which these oligomeric lectins drive narrow membrane bending has been termed the glycolipid-lectin (GL-Lect) hypothesis[4,9–12]. For the cellular uptake of glycosylated cargo proteins such as the cell adhesion molecules CD44 and α5β1 integrin, the current model is as follows: once non-conventionally secreted in an autocrine or paracrine manner into the extracellular milieu[13], the well-explored galectin-3 (Gal3) binds at the cell surface as a monomer to the carbohydrate groups of cargo proteins, and then oligomerizes[7,14]. Oligomerized Gal3 gains the capacity to interact with glycosphingolipids (GSLs) in a way such as to drive the formation of deep and narrow membrane invaginations, both on model membranes and in cells[7]. Thereby, tubular endocytic pits are formed from which clathrin-independent carriers (CLICs) are generated.

Convergent evidence demonstrates that the GL-Lect mechanism is also used by GSL-binding Shiga toxin[6], cholera toxin, simian virus 40 (ref. [15]), and norovirus[16], the GPI-anchored protein CD59 (ref. [8]), and the immunoglobulin superfamily member CD166 (ref. [17]). For some of these, a localization to CLIC structures has been documented[18,19]. However, the physiological contexts in which the GL-Lect mechanism operates in vivo have remained unexplored.

UDP-glucose ceramide glucosyltransferase, encoded by the *Ugcg* gene, is an essential enzyme that catalyzes the first step of GSL biosynthesis[20]. In the inducible *Ugcg*flox/Cre-villin mouse model, the enzyme is genetically removed in a spatiotemporally controlled manner, to acutely deplete GSLs from mouse enterocytes[21]. Using this model, it was clearly shown that GSL content in intestinal tissue is crucial for viability, since animals die within a few days after *Ugcg* depletion, mainly because of malnutrition caused by a failure of nutrients uptake[21]. Since GSLs are key players in the GL-Lect hypothesis, we reasoned that one of the physiological functions of this mechanism might be in lipid and nutrient absorption.

Enterocytes are the most abundant cells lining the intestine. Their contribution to nutrient uptake is well established, notably in the small intestine[22]. Examples are fat-soluble vitamins[23], the iron transporter transferrin[24,25], and the glucose-clearance hormone insulin[26]. All these are internalized by endocytosis from the apical membrane that faces the intestinal lumen and then released after intracellular trafficking on the opposing basolateral side to reach the bloodstream. This process is known as transcytosis, for which the macromolecular neonatal Fc receptor is amongst the best-studied examples (reviewed in ref. [27]).

Lactotransferrin (LTF; also termed lactoferrin) from breast milk and mucosal secretions has been involved in iron transport across the intestinal lining, notably during the early stages of life[28,29]. LTF binds iron with high affinity and at a broad pH range, which is favorable for its function under the slightly acidic conditions of the small intestine[30]. Two LTF receptors have been identified: intelectin-1, also known as omentin or intestinal lactoferrin receptor[31], and moonlighting glycolytic enzyme glyceraldehyde-3-phosphate dehydrogenase[32,33]. Of note, other iron transport mechanisms appear to be able to substitute for LTF, as iron absorption in infants was found to be even increased

after depletion of LTF from mother milk[34], and LTF knockout (KO) mice have a normal iron status[35]. Furthermore, LTF has been described to have immune-modulatory functions and anti-microbial, antiviral, antioxidant, anticancer, and anti-inflammatory activities[30]. Interestingly, studies on Caco-2 cells indicated that LTF is taken up from the apical membrane and undergoes transcytotic trafficking[36,37].

In this study, we have analyzed the involvement of key molecules of the GL-Lect mechanism in endocytic trafficking at the level of the enterocytes of the small intestine of mice. We have found that Gal3 and its newly identified interacting partner LTF are both transcytosed from the apical to the basolateral surface of enterocytes in the intact jejunum. Endocytosis of Gal3 and LTF was strongly inhibited upon inducible depletion of GSLs, and LTF was itself dependent on Gal3 expression to be efficiently transcytosed across enterocytes. In ultrastructural experiments, Gal3 was found in morphologically defined CLICs. These data collectively demonstrate that the GL-Lect mechanism is operating in enterocytes for polarized transcytotic trafficking from the apical to the basolateral membrane.

## Results

### Gal3 is expressed within the intestinal epithelium of the jejunum.
The endocytic phenotype of GSL-depleted enterocytes[21] incited us to test whether the GL-Lect mechanism was operating in these cells. We focused our attention on the jejunum of the small intestine of C57BL/6 mice, which has developed a morphological organization into villi and crypts (Fig. 1a). To identify galectins that are expressed in this tissue, cells of the intestinal mucosa were collected, lysed, and galectin content was enriched via pull-down using a lactose affinity column (lactose-coated beads, see "Methods" for details). The following galectins were detected using mass spectrometry (MS): Gal1, Gal2, Gal3, Gal4, Gal8, and Gal9 (Fig. 1b), consistent with published findings[38].

We focused our attention on Gal3, which in the context of the GL-Lect hypothesis is the galectin that is best studied[7]. To analyze and validate the expression profile and localization of endogenous Gal3 in enterocytes of the jejunum, we performed immunofluorescent staining using anti-Gal3 antibodies. In wild-type C57BL/6 mice, Gal3 expression in enterocytes was detected in the cytoplasm with subapical and basolateral localization patterns (Fig. 1c, arrows). The same experiment was performed on C57BL/6 Gal3 KO mice[39], where no labeling was observed, confirming the specificity of the Gal3 signal (Fig. 1c).

Previous studies had shown that Gal3 is expressed at the surface of epithelial cells in the gastric mucosa and that the protein is abundantly secreted into the mucus layer[40]. However, in our previous immunostaining assay, no Gal3 signal was detected at the more apical region of enterocytes where the mucosa is located (Fig. 1c), likely because PFA fixation did not preserve the mucus layer during sample preparation. To address this point, we used Carnoy's fixative instead. The immunofluorescent analysis demonstrated abundant Gal3 labeling at the apical surface of wild-type C57BL/6 mice, overlapping with the mucus marker UEA-1 (Fig. 1d), confirming that Gal3 was indeed largely localized within the mucus. Tissue from Gal3 KO mice (Gal3−/−) again served as specificity control (Fig. 1d).

### Endocytosis of exogenous Gal3 in enterocytes of the jejunum.
To study the endocytic uptake of Gal3 into enterocytes of the jejunum, the intestine was removed from mice, and recombinant purified fluorophore-labeled Gal3 was injected at 20 µg/mL into the lumen. For these experiments, the intestinal mucus layer needed to be permeabilized to allow exogenous Gal3 to have access to the apical epithelium membrane. Mucus indeed

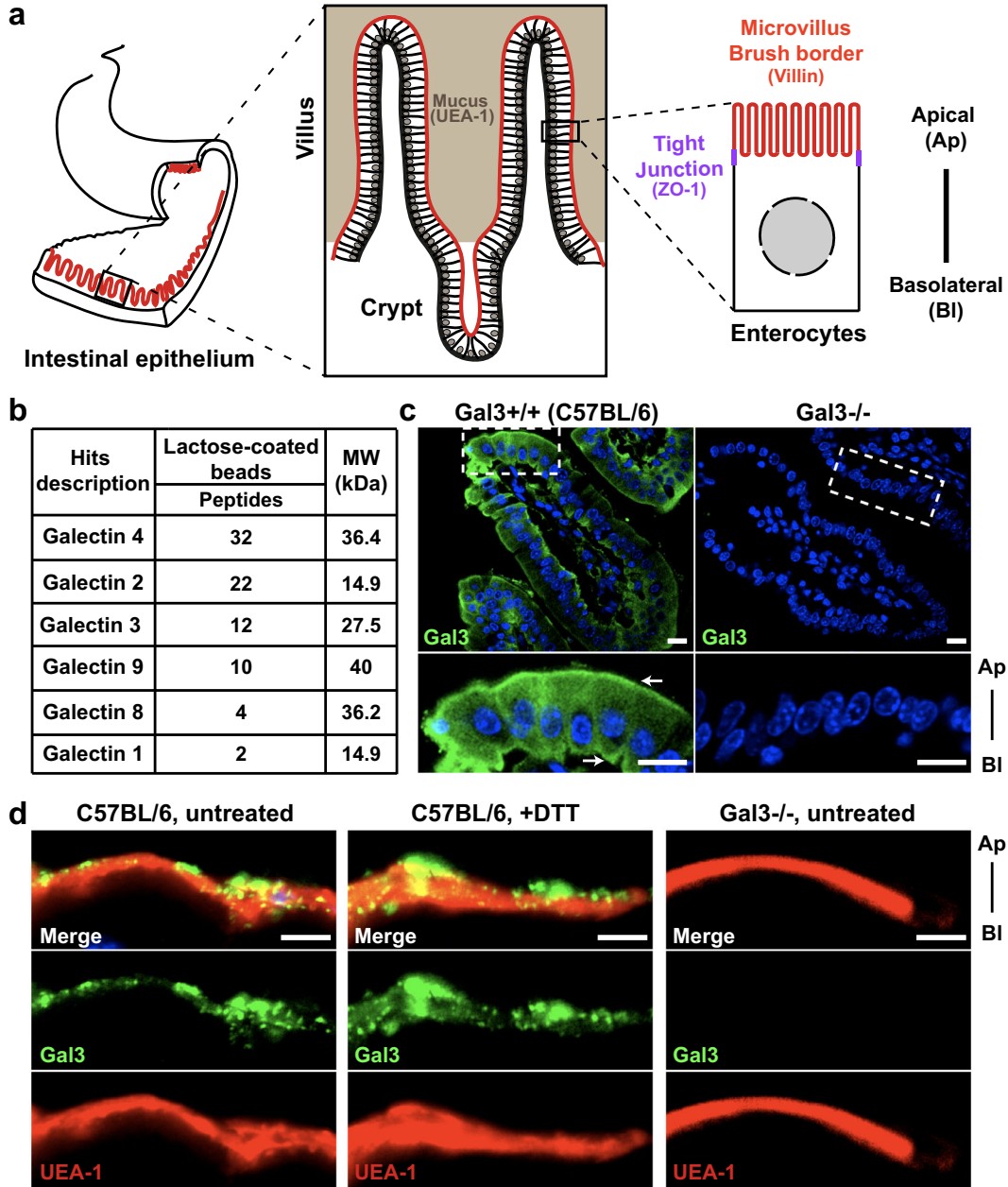

**Fig. 1 Galectin expression in murine jejunum. a** For our studies, the jejunum part of the mouse intestine was used as a model system. Schematic overview of the functional structure and features of the intestinal epithelium and of enterocytes. **b** Galectin expression profile in the jejunum. MS results of galectins that were purified from jejunum cell lysates of C57BL/6 mice, using lactose-coated beads. **c** Localization of endogenous Gal3 (green). Transverse sections of Gal3+/+ (C57BL/6) mouse jejunum were analyzed by immunohistochemistry using an anti-Gal3 antibody. Gal3-deficient (Gal3−/−) mice were included as specificity controls. Insets represent the magnification of a monolayer of cells. Nuclei were labeled with DAPI (blue). Scale bars: 10 μm. **d** Gal3 localization within the mucosal barrier. Immunohistochemistry was performed using sections from Carnoy's fixed jejunum tissue obtained from Gal3+/+ (C57BL/6) or Gal3−/− mice. Labeling with antibodies directed against Gal3 (green) or the mucus marker UEA-1 (red). The absence of Gal3 signal in Gal3−/− conditions confirmed the specificity of the labeling. Note that DTT treatment did not perturb Gal3 or UEA-1 localizations. Scale bars: 5 μm.

prevents inflammation in the digestive tract by forming a barrier that seals the paracellular space and connects individual epithelial cell membranes[41]. Mucus permeabilization was achieved by incubation of the intestinal tissue with dithiothreitol (DTT). Under DTT conditions and using Carnoy's fixative, endogenous Gal3 signal was still observed at the levels of the mucus (Fig. 1d, C57BL/6, +DTT condition). This showed that the mucus was permeabilized, but not removed, and that endogenous Gal3 was retained, likely in interaction with highly O-glycosylated proteins of the mucus.

A number of controls were performed with or without DTT to ascertain that DTT treatment did not damage epithelial integrity: Immunolabeling for the tight junction marker ZO-1 was indistinguishable (Fig. 2a); the ultrastructure of apically localized electron-dense tight junctions was preserved (Fig. 2b) and the brush border marker villin was evenly distributed (Fig. 2c). We also examined tissue permeability under DTT treatment and found that upon incubation at 4 °C, the fluid-phase marker 10-kDa dextran (Fig. 2d) or exogenously added Gal3 (Fig. 2e) did not leak into the intestinal tissue, documenting that the tight junctions were still intact.

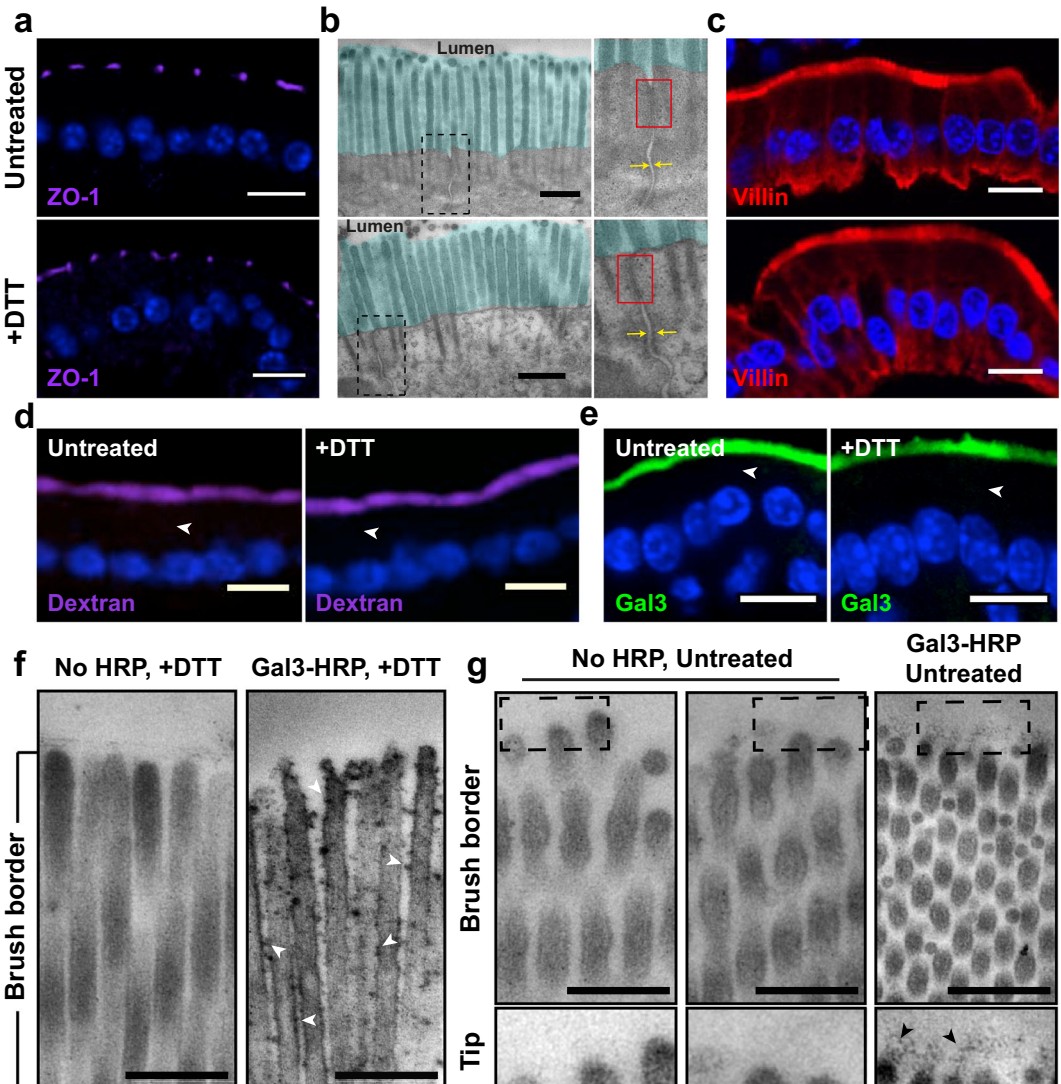

**Fig. 2 Interaction of exogenous Gal3 with murine jejunum. a** Characterization of the integrity of the DTT-treated mouse jejunum. Apical tight junctions, as visualized using ZO-1, are not altered under DTT treatment conditions. **b** Electron-dense tight junctions (red rectangles) have unperturbed morphology and localization in DTT treatment conditions, and the cohesiveness of lateral cell–cell contacts (yellow arrows) are well preserved, as analyzed by electron microscopy. The turquoise blue highlights the microvilli (brush border) zone. **c** The microvilli marker villin was used to label the apical integrity of the epithelium. Note that villin and ZO-1 are both normally localized under DTT treatment conditions. Nuclei in blue. **d** 10-kDa dextran was incubated at 4 °C with DTT-treated murine jejunum to analyze tissue permeability, which was not altered under DTT treatment conditions. **e** Gal3 was incubated at 4 °C with DTT-treated murine jejunum. As with 10-kDa dextran, no transcellular leakage was observed under DTT treatment conditions (white arrowheads). Nuclei in blue. **f** DTT treatment is required for Gal3 to reach the brush border. DTT-permeabilized jejunum that was incubated with Gal3-HRP shows HRP-positive signal at the level of microvilli (dark dotted signals, white arrowheads). This labeling is specific as it was not observed when incubations were done in the absence of Gal3-HRP (No HRP). **g** Without DTT treatment, Gal3-HRP signal is trapped within the mucus layer. Granular dark dotted structures are detected at the far tip part of the microvilli, further visible under higher magnification (right image and zoom, black arrowheads). In contrast, no signal is detected in the absence of Gal3-HRP (two left images and zooms). Scale bars = 10 μm for immunofluorescence, and 1 μm for electron microscopy images.

Access of Gal3 to microvilli was then analyzed by electron microscopy. The localization of horseradish peroxidase (HRP)-labeled Gal3 to microvilli of the enterocytes could be detected after mucus permeabilization with DTT (Fig. 2f; see arrowheads to indicate HRP-catalyzed precipitates on the surface of microvilli). No precipitate was observed in the absence of Gal3-HRP (Fig. 2f, no HRP condition), which documented the specificity of the signal. In the absence of DTT treatment, dark granular DAB precipitates from Gal3-HRP were only detected above the apical microvillar membrane (Fig. 2g), likely because the proteins remained trapped in the mucus layer. The absence of Gal3-HRP again served as specificity control (Fig. 2d, no HRP condition). These ultrastructural studies thus clearly

established that exogenous Gal3 reached the apical membrane of small intestinal enterocytes only after DTT-mediated mucus permeabilization.

To assess the Gal3 internalization profile, exogenous fluorophore-labeled Gal3 (Gal3-488) was bound at 4 °C to the apical surface of enterocytes in DTT-permeabilized intestinal tissue, after which the tissue was washed and shifted to 37 °C to allow for internalization. After the indicated periods of time, the intestinal tissue was placed on the ice again, and Gal3 that had not yet been internalized (i.e., that was still cell surface accessible) was removed by incubation with a 200 mM lactose wash solution. Therefore, Gal3 labeling that is visible in Fig. 3a represents internalized molecules. Already after 5 min of incubation at 37 °C,

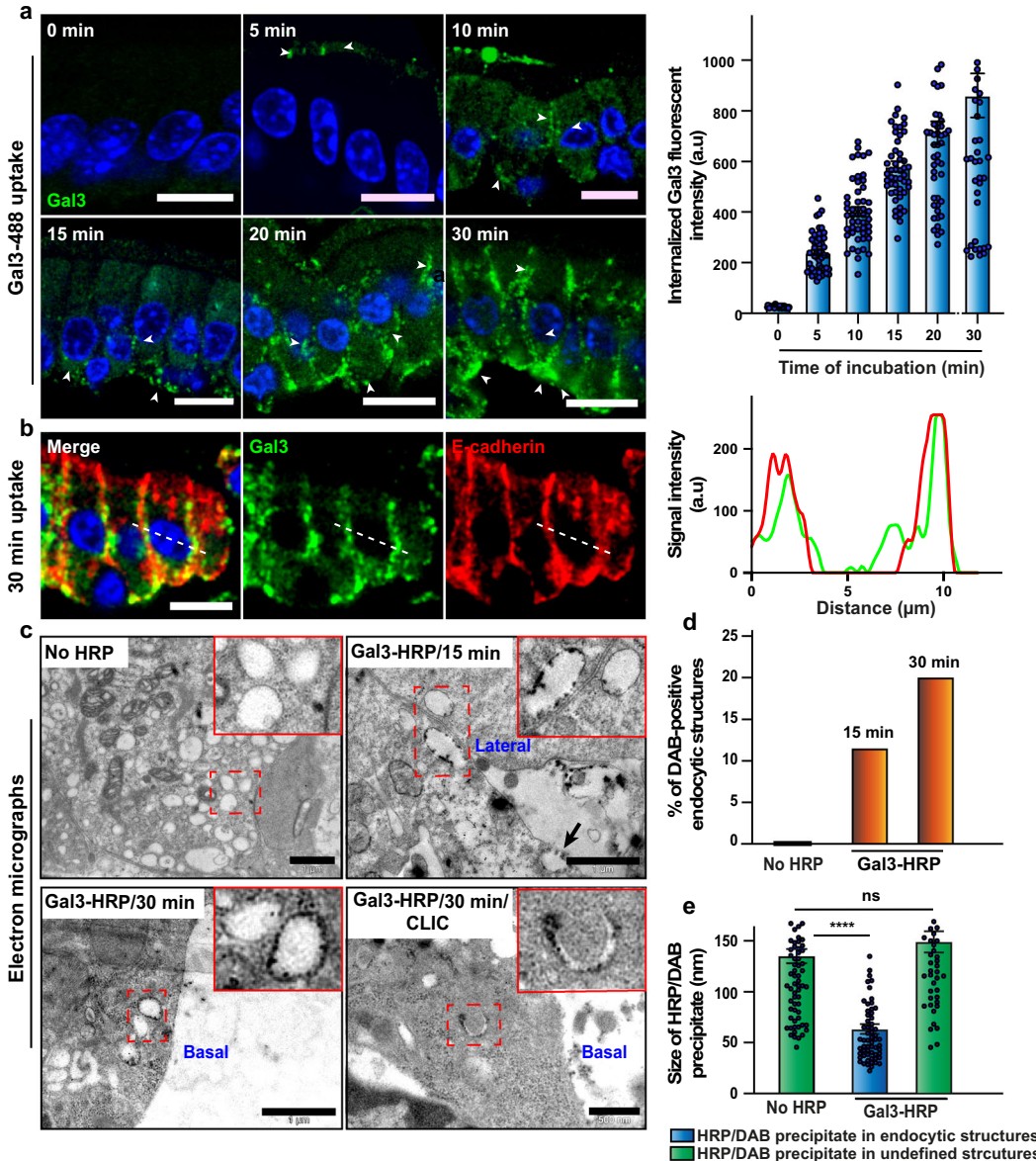

**Fig. 3 Internalization of exogenous Gal3 into enterocytes. a** Pulse-chase experiment of Gal3 uptake in enterocytes. Gal3 (green) was bound at 4 °C to enterocytes of DTT-treated jejunum. After washing, incubations were performed at 37 °C for the indicated periods of time. Cell surface-exposed Gal3 was removed with lactose wash. Note that no signal is detected when cells were not incubated at 37 °C (0 min). However, when shifted to 37 °C, distinct intracellular structures become visible as early as after 5 min of incubation (arrowheads). The fluorescence signal of internalized Gal3 is shown to the right in function of time (means ± SEM, $n = 50$ cells quantified per condition, one representative of three independent experiments). Nuclei in blue. Scale bars = 10 μm. **b** Apically internalized Gal3 accumulates at the basolateral membrane. Enterocytes were continuously incubated for 30 min at 37 °C with apically added Gal3 (green). Cell surface-exposed Gal3 was removed by lactose wash. Cells were labeled for the adherens junction marker E-cadherin, which is known to be localized to the basolateral membrane. A clear overlap between Gal3 and E-cadherin (RGB plot) indicates that Gal3 is indeed transcytosed in these enterocytes. Nuclei in blue. Scale bars = 10 μm. **c** Electron microscopy analysis of Gal3 uptake into enterocytes of the DTT-treated jejunum. After 15 min of incubation at 37 °C with 40 μg/mL of HRP-coupled Gal3, electron-dense DAB precipitates (arrowheads) are found in vacuolar structures close to the lateral membrane (Gal3-HRP/15 min micrograph) and mainly accumulate at the basal side after 30 min of incubation (Gal3-HRP/30 min micrograph). Some tubular crescent-shaped structures could be detected with typical CLIC morphology (Gal3-HRP/30 min/CLIC micrograph). For comparison, a cell that was incubated in the absence of Gal3-HRP is shown in the No HRP micrograph. Red-dashed insets represent endocytic structures. Scale bars = 500 nm for the bottom-right electron microscopy image and 1 μm for the others. **d** Endocytic structures containing electron-dense DAB precipitate were quantified in the indicated conditions. Percentage of DAB-positive endocytic structures (no HRP condition, $n = 525$ endocytic carriers; Gal3-HRP/15 min condition, $n = 224$ endocytic carriers; Gal3-HRP/30 min condition, $n = 131$ endocytic carriers were analyzed). **e** Two types of precipitates were detected, of which only the smaller one was specific for the Gal3-HRP condition. Quantification of the size of dark precipitates (means ± SEM; no HRP condition, $n = 79$ precipitates were measured within undefined structures; Gal3-HRP condition, $n = 62$ precipitates were measured within endocytic structures, and $n = 49$ precipitates in undefined structures). Ordinary one-way ANOVA, ****$P < 0.0001$, ns non-significant.

such internalized Gal3 labeling could be detected in subapical localizations (Fig. 3a, arrowheads). After 10 min of incubation at 37 °C, we observed Gal3-containing dotted structures mainly at the lateral side of the cell, which then started to be located more basally after 15 min of internalization. After 20–30 min, the Gal3 signal became most prominent at the opposite basolateral domain (Fig. 3a), overlapping with the basolaterally located adherens junction marker E-cadherin (Fig. 3b, line scan to the right from the white dotted line on the image). The evolution of the labeling pattern was paralleled by increased intracellular accumulation of Gal3 (Fig. 3a, histogram). These results clearly document a gradual transcytotic trafficking pattern of Gal3 from the initial site of binding at the apical membrane of enterocytes to the basolateral membrane.

Electron microscopy was used to analyze the morphology of Gal3-containing endocytic structures at the ultrastructural level. For these experiments, we again worked with HRP-coupled Gal3, which retains the intracellular trafficking characteristics of non-modified Gal3 (ref. [7]). After incubation of 40 µg/mL of Gal3-HRP at 37 °C with DTT-permeabilized intestinal tissue, electron-dense diaminobenzidine (DAB) precipitates were found in vacuolar structures of 200–400 nm in diameter that were mostly located laterally after 15 min (Fig. 3c, Gal3-HRP/15 min micrograph), and at the basal side after 30 min (Fig. 3c, Gal3-HRP/30 min micrograph), kinetics that was consistent with the immunofluor-escence experiments of Fig. 3a. These vacuolar structures were devoid of electron-dense precipitates on cells to which Gal3-HRP had not been added (Fig. 3c, no HRP micrograph), which documented the specificity of the signal. The number of DAB-positive endocytic structures increased with incubation time (Fig. 3d), as expected. Larger electron-dense signals were found in diffuse localization under all conditions, i.e., even in the absence of Gal3-HRP (Fig. 3e, green columns), which were therefore considered as background.

Of note, some Gal3-HRP/DAB-positive structures were visible in the close vicinity of the lateral membrane, and sometimes apparently fusing with it (Fig. 3c, Gal3-HRP/15 min micrograph, arrow), in agreement with a basolateral fate for internalized Gal3. Furthermore, we also observed Gal3-HRP/DAB-positive 300–600 nm tubular and crescent-shaped structures, which had the typical morphology of CLICs (Fig. 3c, Gal3-HRP/30 min/CLIC micrograph).

**Gal3 transcytosis is GSL-dependent.** According to the GL-Lect hypothesis, oligomeric Gal3 interacts with GSLs to induce narrow membrane bending for the construction of tubular endocytic pits[9]. To test the functional implication of GSLs in Gal3 endocytosis and transcytosis in mouse enterocytes, we took advantage of an inducible UDP-glucose ceramide glucosyltransferase (encoded by the Ugcg gene) KO mouse model to deplete GSLs specifically in these cells[21]. To analyze GSL expression levels on the first 3 days after tamoxifen (TAM) injection, enterocytes were scraped off the lamina propria of the jejunum, GSLs were extracted and analyzed by thin-layer chromatography (TLC) (Fig. 4a), or MS (Fig. 4b). We focused our analysis on the hexa-ceramide glucosylceramide (GlcCer), the initial product of UGCG, and asialo GM1 (GA1), both of which are major GSLs of adult enterocytes[21]. Using these analytical methods, we found that in the Ugcgflox/Cre+ genetic background, both GSLs were efficiently and significantly depleted already at day 1 post TAM injection (55/70% decrease of GA1/GlcCer, respectively), which further dropped by day 2 (80/72% GA1/GlcCer decrease) and day 3 (83/82% GA1/GlcCer decrease) (Fig. 4a, b). Expectedly, expression levels of the ceramide-containing but glycosphingolipid-unrelated sphingomyelin (SM), whose synthesis is not dependent on the UGCG enzyme, were not

significantly perturbed (Fig. 4b), which showed that lipid expression was not generally affected.

GSLs are major lipids of the membranes of the intestinal epithelium, and key players for the maintenance of apicobasal polarity[21,42]. It was therefore of outstanding importance to characterize the integrity of epithelial features at different days of GSLs depletion. The apical localization pattern of villin remained unchanged at days 1 and 2 post-TAM induction (Fig. 4c, -GSLs; quantified in Fig. 4d). Note that GlcCer and GA1 levels were already strongly decreased at these time points (Fig. 4b). On day 3 post-TAM induction, the villin signal was hardly detectable anymore (Fig. 4c, -GSLs; quantified in Fig. 4d). Similarly, the tight junction protein ZO-1 was correctly localized at days 1 and 2 post-TAM induction and strongly mislocalized at day 3 (Fig. 4e). Taken together, these experiments demonstrated that the overall integrity of the enterocyte tissue remained unperturbed up to 2 days post-TAM-induced GSL depletion.

The mucosa of TAM-treated, GSL-depleted intestinal tissue was permeabilized with DTT, and incubated at 4 °C with 20 µg/mL of Gal3-488, followed by washing. Gal3 binding to cells was similar at days 1 and 2 after TAM injection but dramatically dropped to background levels at day 3 (Fig. 5a). Unperturbed binding at days 1 and 2 is consistent with previous findings on cells in culture in which it was also shown that GSLs were not required for the interaction of Gal3 with cells[7]. In contrast, when incubation with Gal3 was done continuously at 37 °C, its endocytic uptake was already reduced by half on day 1 after TAM-induced GSL depletion and was further decreased by 65% or 80% on days 2 or 3, respectively (Fig. 5b). This increasing inhibition of Gal3 endocytosis was closely correlated to decreasing levels of GSLs under corresponding conditions (Fig. 4a, b), strongly suggesting that both were causally linked.

We next analyzed the contribution of the clathrin machinery in the transcytotic process of Gal3, using the antibiotic ikarugamy-cin (Ika), which acutely inhibits clathrin-mediated endocytosis by disrupting coated pit formation[43]. Upon Ika treatment, the endocytic uptake of Gal3 was only mildly reduced (29%) (Supplementary Fig. 1a), consistent with previous observations on cells in culture[7]. As a positive control, we have assessed the endocytosis of a pentameric immunoglobulin M (IgM), for which it has been reported that internalization mainly occurs in a clathrin-dependent and glycan-independent manner in interaction with Fc receptors[44,45]. The IgM that we have used was directed against the GSL Gb3, onto which no efficient binding is observed in the mouse intestine[46]. It could therefore be assumed that IgM interaction with the epithelium occurred via Fc receptors, and indeed, its internalization was 92% decreased upon Ika treatment (Supplementary Fig. 1b). Taken together, these findings confirm that the transcytotic uptake of Gal3 relies mainly on a clathrin-independent mechanism.

A few more endocytic cargoes were used to further characterize the endocytic trafficking in the enterocytes of the jejunum of mice. The overall endocytic activity of the intestinal tissue was tested using a bulk fluid-phase marker, 40-kDa dextran, which enters into the cells via all available micro and macropinocytosis processes[47] to accumulate into typical supranuclear vacuolar-shaped lysosomes. 40-kDa dextran endocytosis was unperturbed for up to 2 days after TAM-mediated GSL depletion (Fig. 5c) and expectedly dropped at day 3 (Fig. 5c) when the intestinal tissue gets disorganized (see ref. [21] and above). It thereby appears that in primary enterocytes of murine jejunum, endocytic processes that operate according to the GL-Lect hypothesis are contributing a minor fraction to overall fluid-phase uptake. The clathrin-dependent endocytic uptake of IgM was also not affected by GSL depletion (Supplementary Fig. 2a), which again established

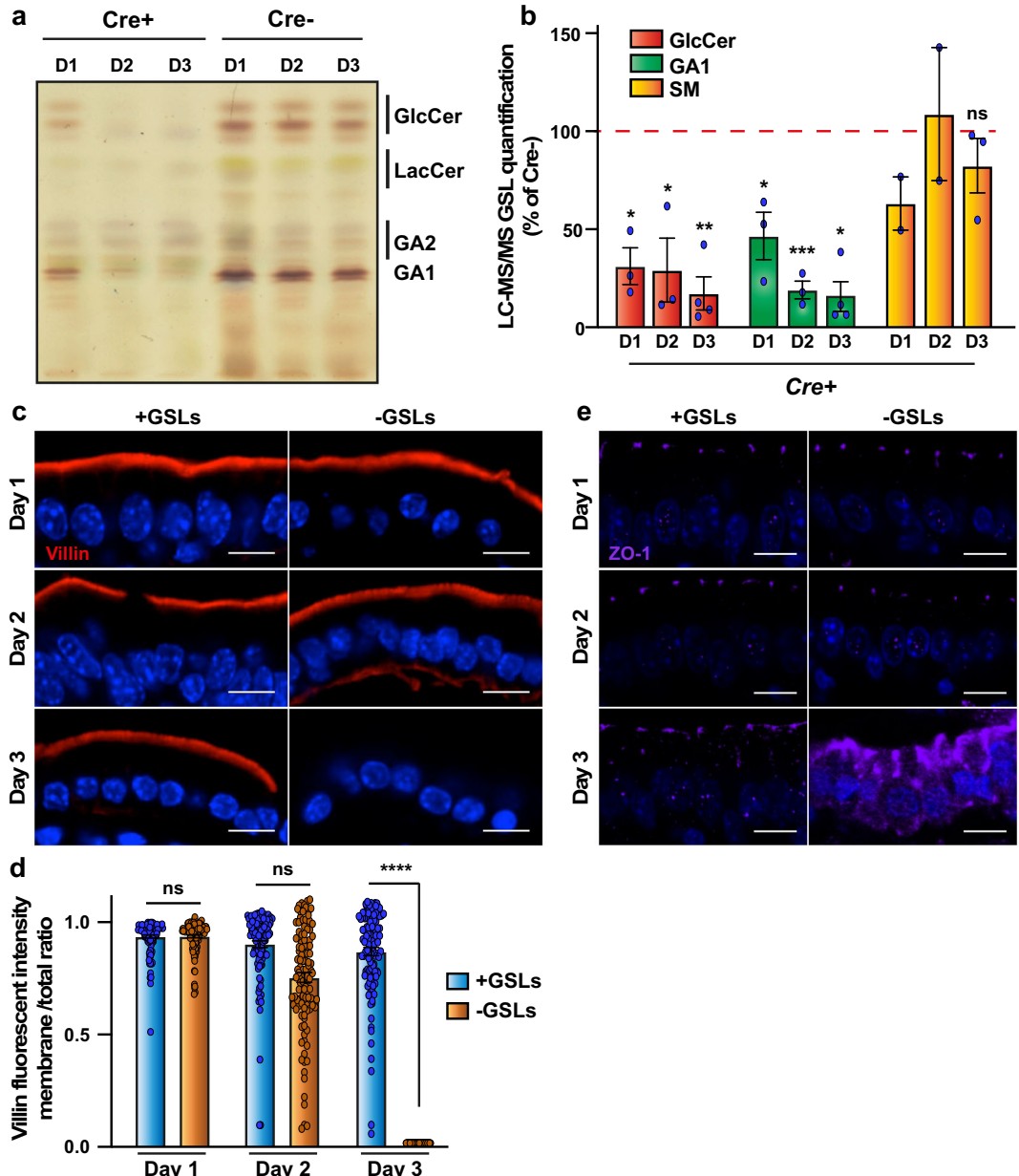

**Fig. 4 Tissue integrity under GSL depletion conditions. a** *Ugcg*flox/Cre[+] or *Ugcg*flox/Cre[−] mice were injected with 1 mg of TAM. At days 1, 2, or 3 post-injection, the jejunum was excised and analyzed for GSL expression using thin-layer chromatography. D1-3(Cre−) samples contain a GSL-triple-band migrating at the height of the GlcCer standard, and a double band at the height of the GA1 standard (indicated on the right side). Non-purple bands are other lipids than GSLs. All GSL bands are strongly reduced in the D1-3(Cre+) samples. GA2 = Gg$_3$Cer, GA1 = Gg$_4$Cer. **b** Relative quantification of GSL by LC-MS/MS. Lipid extracts containing internal standards were analyzed by RP18-LC-MS/MS. Total intensities of GSL signals were normalized to the internal standards and to the intensity of the corresponding untreated control samples (% of Cre−). The GSL-unrelated sphingomyelin did not display any significant alteration upon *Ugcg* depletion. Mean percentage ± SEM. GA1/GlcCer quantification, three independent experiments for D1 and D2, and 4 independent experiments for D3; SM quantification, two independent experiments for D1 and D2, and three independent experiments for D3. Student's unpaired *t* test, *$P < 0.03$, **$P < 0.002$, ***$P < 0.0002$, ns: non-significant. **c–e** Conservation of tissue integrity under GSL depletion conditions. Immunohistochemistry on jejunum of *Ugcg*flox/Cre[−] (+GLSs) or *Ugcg*flox/Cre[+] (-GSLs) mice at the indicated days after TAM injection. Immunolabeling for villin (**c**) or ZO-1 (**e**). Note that at days 1 and 2 after TAM injection, staining patterns are not visibly altered under GSL depletion conditions. Data from (**c**) are quantified in (**d**) (membrane signal/total signal ratio ± SEM, $n = 113$ cells quantified for day 1 *Ugcg*flox/Cre[−] and day 2 *Ugcg*flox/Cre[+], and $n = 112$ cells quantified for the other conditions; one representative of three independent experiments). Nuclei in blue. Student's unpaired *t* test, ****$P < 0.0001$, ns: non-significant. Scale bars = 10 μm.

the specificity of the observed GSL depletion phenotype on Gal3 uptake and transcytosis (see Fig. 5b).

CD59 is a GPI-anchored protein whose endocytic uptake was shown to be clathrin-independent in cells in culture[8]. This also seems to be the case in primary enterocytes of murine jejunum, as

inhibition of clathrin machinery by Ika did not alter CD59 endocytosis (Supplementary Fig. 1c). CD59 accumulated in a subapical punctate pattern, likely representing recycling endosomes[48]. In contrast to Gal3, CD59 internalization into enterocytes of *Ugcg*flox/Cre[+] mice also did not show any major perturbation

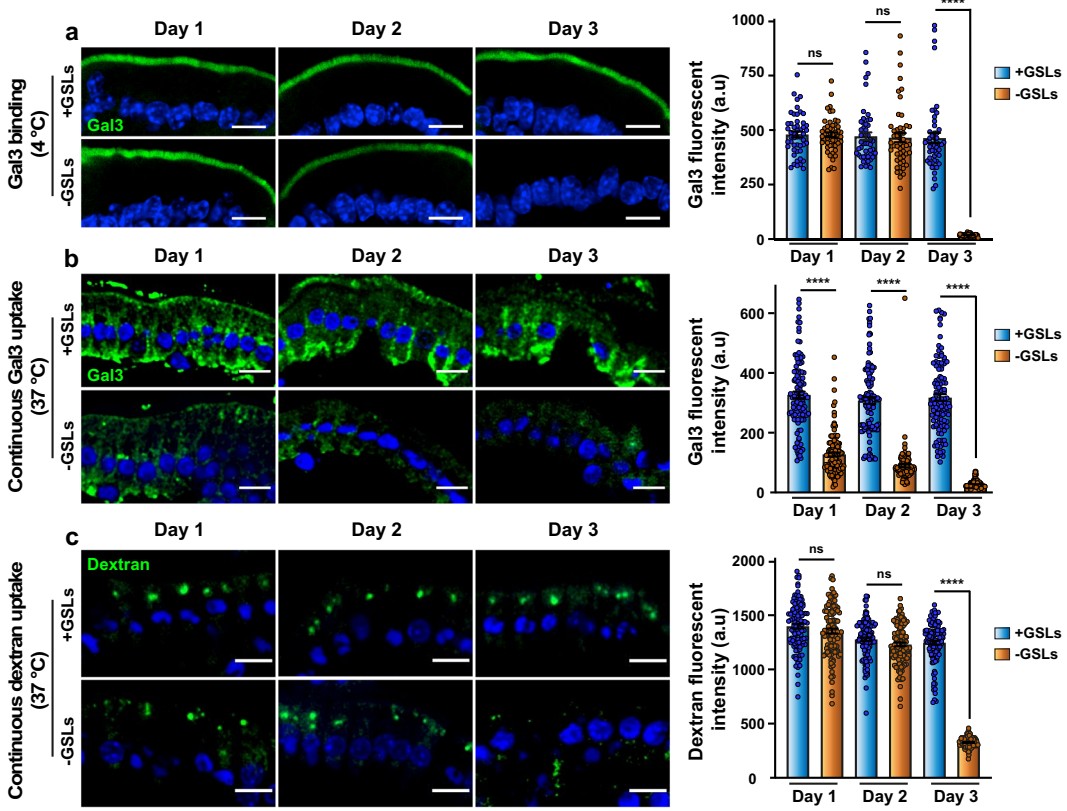

**Fig. 5 Gal3 undergoes GSL-dependent endocytosis. a** DTT-treated jejunum of TAM-injected *Ugcg*flox/Cre[+] (-GSLs) or *Ugcg*flox/Cre[-] (+GLSs) mice was incubated at 4 °C with 20 μg/mL of Gal3 (green). Note that Gal3 binding to the apical surface was similar at days 1 and 2 after TAM injection, while it dropped to background levels at day 3. Right: Quantification of apical signal (means ± SEM, $n = 50$ cells analyzed per condition, one representative of four independent experiments). **b** Experiment as in (**a**), except that incubation with Gal3 (green) was done for 30 min at 37 °C. Cell surface-exposed Gal3 was removed with lactose wash. Note that Gal3 uptake was strongly inhibited in GSL-depleted enterocytes, starting with day 1. Right: Signal quantification (means ± SEM, $n = 114$ cells per condition, one representative of four independent experiments). **c** Experiment as in (**b**), except that incubation at 37 °C was performed with 1 mg/mL of 40-kDa dextran (green). Dextran uptake was similar at days 1 and 2 after TAM injection, while it dropped to background levels at day 3. Right: Representative signal quantification (means ± SEM, $n = 109$ cells per condition, one representative of four independent experiments). Nuclei in blue. Statistical analysis: Student's unpaired *t* test, ****$P < 0.0001$, ns: non-significant. Scale bars = 10 μm.

for up to 2 days after TAM-mediated GSL depletion (Supplementary Fig. 2b). Since evidence from cell culture experiments suggests a dependency of CD59 internalization on galectins[8], it might be speculated that CD59 itself provides the glycolipid aspect for its uptake according to the premises of the GL-Lect hypothesis.

**Transcytosis of the Gal3 interactor LTF depends on GSLs.** Up to now, we have seen that the GL-Lect driver Gal3 undergoes GSL-dependent endocytosis and transcytosis in enterocytes of the murine jejunum. To identify cargo proteins whose trafficking in enterocytes would be regulated by Gal3, we performed pull-down experiments. DTT-treated jejunum was incubated for 30 min at 4 °C with 20 μg/mL of His-tagged Gal3. Enterocytes were scraped off the lamina propria, lysed, and Gal3-binding proteins were identified by mass spectrometry after Gal3-His immobilization onto cobalt–agarose beads. In addition to other proteins, LTF showed up as one of the potential Gal3-binding partners with a systematic occurrence (Fig. 6a). To confirm this interaction, purified human Gal3-Cy3 and mouse His-tagged LTF were incubated together. LTF-(His) was pulled down via cobalt–agarose beads, and elution was done either with denaturing sample buffer to remove all proteins from beads, or with lactose to specifically break the interaction between Gal3 and the sugar groups of bead-immobilized LTF-(His). Eluates were analyzed by SDS-PAGE. Gal3 was clearly co-immunoprecipitated with LTF (Fig. 6b and 1st

lane, Supplementary Fig. 3). When elution was done with lactose, only Gal3 was found on the gels (Fig. 6b and 3rd lane, Supplementary Fig. 3), which confirmed the specificity of the interaction between both proteins. Furthermore, when incubation was performed in the presence of the I3 compound that acts as a specific inhibitor of Gal3 binding to the carbohydrates of natural glycoproteins[49], the pulled-down amount of Gal3 was visibly reduced (Fig. 6b and 2nd lane, Supplementary Fig. 3), which again confirmed the specificity of this interaction.

Fluorophore-labeled human Gal3 (Gal3-488, green) and mouse LTF (LTF–Cy3, red) were then incubated together for 30 min at 37 °C with small intestinal tissue preparations on which the mucus had been permeabilized by DTT treatment. A strong overlap was observed between both markers within endocytic structures (Fig. 6c, see line profiles), located mainly laterally within cells (Fig. 6c-1, RGB profile), or basally (Fig. 6c-2, RGB profile). The overlap was quantified using Manders' coefficient (Fig. 6d). It could be concluded that Gal3 and LTF showed a similar transcytosis pattern.

The GSL dependency of LTF transcytotic trafficking within DTT-permeabilized enterocytes was determined in experiments that were similar to the ones described above for Gal3. Upon incubation at 4 °C, fluorophore-labeled LTF bound efficiently to the apical membrane of enterocytes from control mice (+GSLs) at all time points after TAM injection (Fig. 6e, +GSLs). Up to 2 days after TAM injection, an unperturbed LTF-binding level

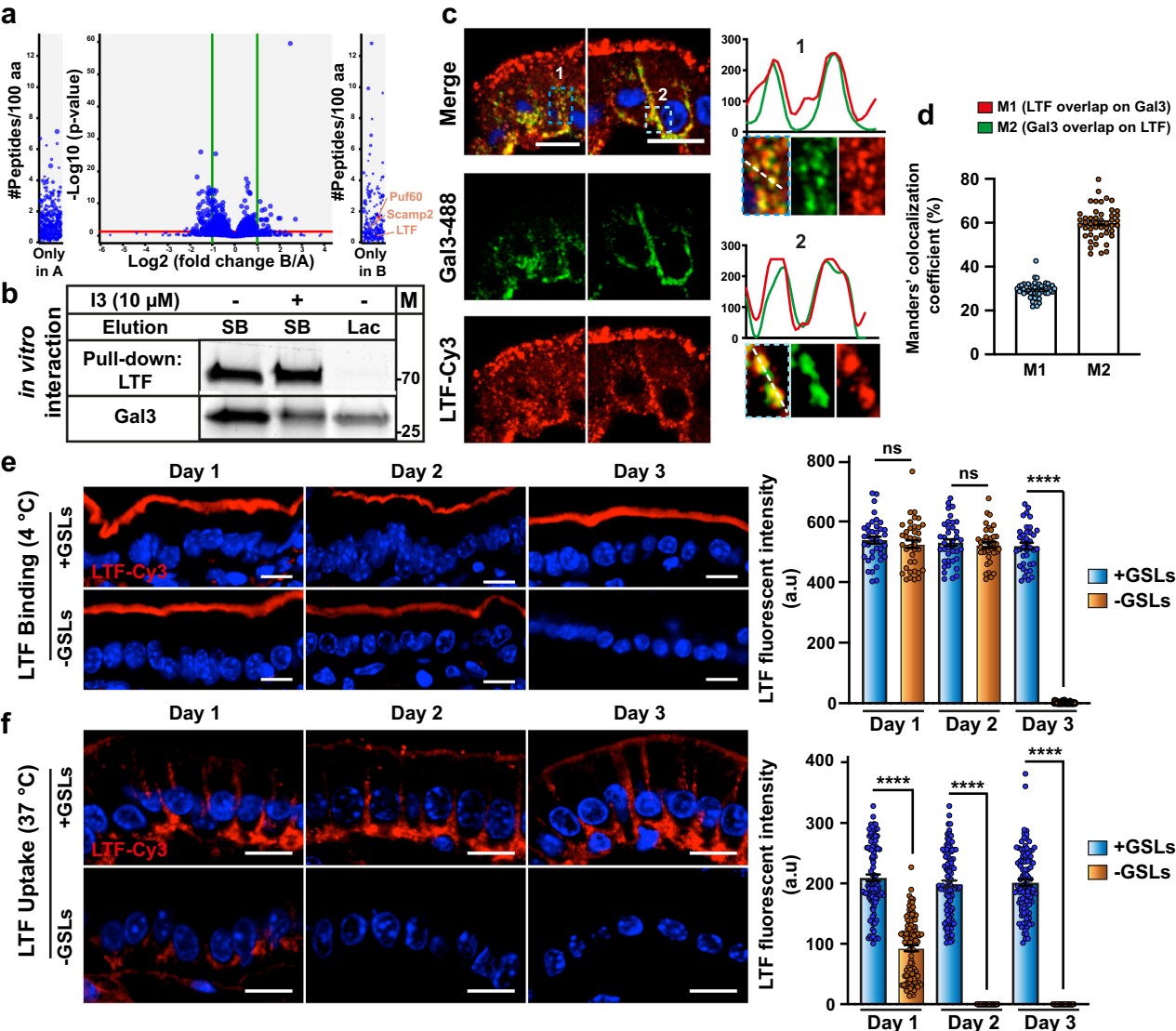

**Fig. 6 LTF transcytosis is GSL-dependent. a** Identification of LTF as a Gal3 interacting protein in enterocytes of the jejunum of wild-type C57BL/6 mice. Top: Quantitative analysis of proteins present in Gal3 (=B) samples compared to PBS control (=A) is shown as a volcano plot. x axis = log2 (fold change Gal3/PBS), y axis = −log10 (P value). The horizontal red line indicates P value = 0.05, vertical green lines indicate absolute fold change = 2. Data represent the results of three independents pull-down experiments. Proteins are shown with at least one peptide in each of the Gal3 replicates, and zero peptide in the PBS control replicates in the qualitative comparison without Match-between-Runs-rescued peptides. **b** Pull-down experiment on cobalt–agarose beads of purified LTF-His and Gal3-Cy3. Note that Gal3 is co-purified with LTF in a Gal3 inhibitor (I3)-dependent manner, and specifically eluted with lactose (one representative of two independent experiments). M = PageRuler™ prestained protein ladder (the position of bands 25 and 70 kDa is shown in the gel). **c** In total, 20 µg/mL of Gal3-488 (green) and LTF–Cy3 (red) were co-incubated for 30 min at 37 °C with DTT-permeabilized jejunum. Note the strong overlap between both markers. **d** Mander's coefficient (%) analysis confirms colocalization between Gal3 and LTF, strongly suggesting that both proteins are co-internalized (mean percentage ± SEM, n = 42 cells). **e** At days 1, 2, or 3 after TAM injection, DTT-permeabilized jejunum of Ugcgflox/Cre⁻ (+GSLs) or Ugcgflox/Cre⁺ (-GSLs) mice was incubated at 4 °C with 20 µg/mL of LTF-Cy3 (red). Note that LTF binding to the apical surface was similar in all conditions, with the exception of day 3 after TAM injection where it dropped to background levels. Right: Quantification of apical signal (means ± SEM, n = 38 cells per condition, 1 representative of 4 independent experiments). **f** Experiment as in (**c**), except that incubation, was performed for 30 min at 37 °C. LTF endocytosis was strongly reduced already at day 1 after TAM injection and dropped to background levels at days 2 and 3. Right: Signal quantification (means ± SEM, n = 102 cells per condition, one representative of four independent experiments). Nuclei in blue. Statistical analysis in (**c**) and (**d**): Student's unpaired t test, ****P < 0.0001, ns: non-significant. Scale bars = 10 µm.

was observed in GSL-depleted (−GSLs) mice, while at day 3, LTF binding was hardly detectable anymore (Fig. 6e, −GSLs; quantification to the right). These findings closely mirror those observed for Gal3 binding under corresponding experimental conditions (Fig. 5a).

Upon incubation for 30 min at 37 °C, LTF could be detected at the basolateral domain of enterocytes of control mice (+GSLs) at days 1, 2, and 3 after TAM injection (Fig. 6f, +GSLs). In contrast,

endocytic uptake and transcytosis of LTF were already strongly decreased 1 day after TAM injection and totally inhibited after 2 or 3 days (Fig. 6f, −GSLs; quantification to the right), again mirroring the results that were observed for Gal3 under corresponding experimental conditions (Fig. 5b). Furthermore, again similar to Gal3, the inhibition of the clathrin machinery by incubation of cells with Ika only had a small impact on LTF endocytosis (Supplementary Fig. 1d).

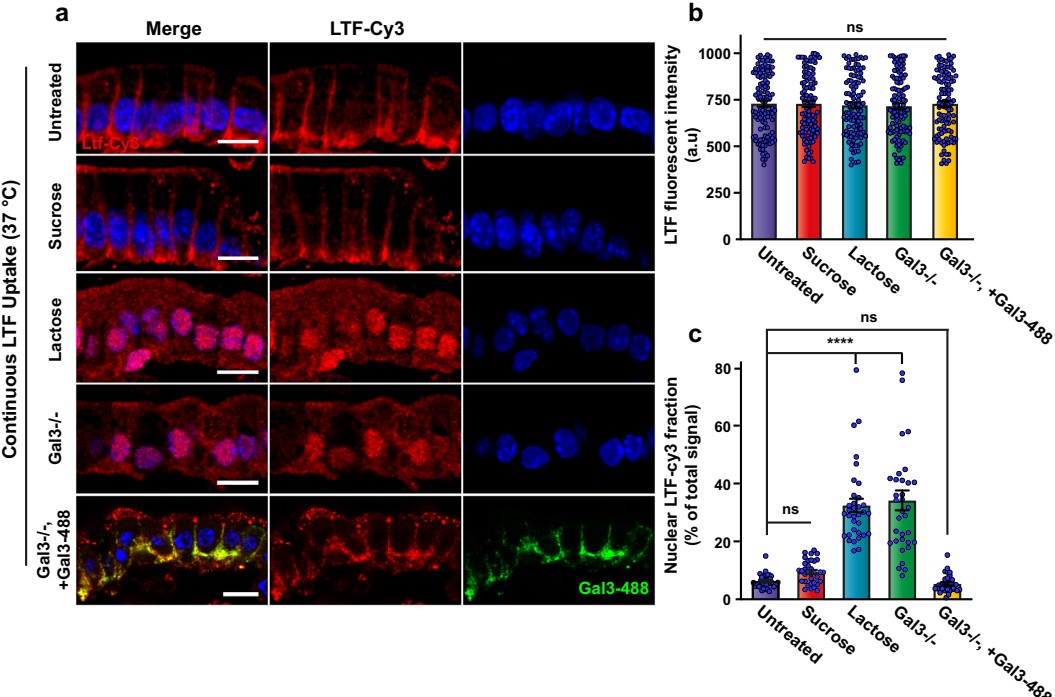

**Fig. 7 LTF transcytosis is Gal3-dependent. a** For all experiments that are shown in this figure, LTF–Cy3 (red) at 20 µg/mL was incubated for 30 min at 37 °C with DTT-permeabilized jejunum. The following experimental conditions are represented: control conditions with wild-type C57BL/6 mice; incubation in the presence of 200 mM sucrose, which did not affect LTF–Cy3 trafficking; incubation in the presence of 200 mM lactose as a competitor for galectin binding to carbohydrates, which strongly reduced the basolateral accumulation of LTF–Cy3; incubation with jejunum from Gal3 KO mice, on which the basolateral accumulation of LTF–Cy3 was strongly reduced; co-incubation of LTF–Cy3 with 20 µg/mL of Gal3-488 (green) on the jejunum from Gal3 KO mice, which led to the rescue of LTF transcytosis to the basolateral surface. **b** Quantification of experiments shown in (**a**) of total LTF–Cy3 labeling intensity per cell in the indicated conditions. Representative quantification (means ± SEM, $n = 108$ cells per condition, one representative of four independent experiments). **c** Quantification of experiments shown in (**a**) of the labeling intensity of LTF in the nuclear area over total LTF intensity (mean percentage ± SEM, $n = 34$ cells per condition, one representative of two independent experiments). Nuclei in blue. Statistical analysis in (**b**) and (**c**): **b** Student's unpaired $t$ test, ****$P < 0.0001$, ns: non-significant; **c** ordinary one-way ANOVA multiple comparisons, ****$P < 0.0001$. Scale bars = 10 µm.

Very clearly, the GSL dependency and clathrin independence of the transcytotic trafficking of LTF mirrored very closely that of its interacting partner Gal3.

**Transcytosis of LTF depends on Gal3.** We then measured LTF uptake under conditions in which the activity of endogenous galectins was inhibited. At high concentrations, lactose acts as a competitive inhibitor of galectin binding to carbohydrates on proteins and lipids. The incubation of LTF with small intestinal tissue in the presence of 200 mM lactose led to a strong reduction of the basolateral accumulation of LTF (Fig. 7a, lactose condition), while LTF uptake and transcytosis were as efficient as under control conditions (Fig. 7a, untreated condition) when lactose was replaced by the disaccharide sucrose, which does not act as a competitor (Fig. 7a, sucrose condition). The genetic deletion of Gal3 (Gal3−/−) also had a dramatic effect on the basolateral accumulation of LTF (Fig. 7a, Gal3−/− condition), a phenotype that was efficiently rescued by the co-incubation of LTF–Cy3 (red) with exogenously added Gal3-488 (green) (Fig. 7a, Gal3−/−, +Gal3 condition).

The total levels of cell-associated LTF were similar in all conditions (Fig. 7b), most likely because LTF enters cells by several endocytic processes (see "Discussion"), of which only the GL-Lect mechanism would couple to sorting towards the basolateral surface via transcytosis. We indeed found that more LTF accumulated in the nuclear region in lactose incubation or Gal3 KO conditions (Fig. 7c), which likely originated from a redirection of the ligand into the nuclear targeting pathway[50].

In conclusion, it became apparent from these studies that endocytosis of Gal3 and LTF in enterocytes of the murine jejunum is GSL-dependent, and that Gal3-dependent uptake of LTF is required for its transcytosis to the basolateral domain.

## Discussion

A likely explanation for the existence of different endocytic mechanisms is that they allow modulating the functions of given cargo molecules. It is therefore of critical importance to identify the physiological contexts in which they operate. In this study, we provide convergent evidence for a critical contribution of the GL-Lect mechanism to transcytosis of LTF in murine enterocytes of the small intestine.

The mucus layer consists of cross-linked glycans that form a functional barrier implicated in inflammatory responses against pathogens. Endogenous Gal3 and other galectins are highly enriched in mucus[51], and in addition to contributing to its stability, they may also form a reservoir from which they can be recruited to build endocytic sites (Fig. 8a). Exogenous Gal3 and LTF, as used in our study, gain access to the apical surface of enterocytes only upon DTT-mediated permeabilization of the mucus layer (Fig. 8a). Using this experimental system, we show that GSL depletion reduces the total amount of internalized and transcytosed Gal3 and LTF, while the inhibition of Gal3 function with competing lactose or genetic deletion of the Gal3 gene specifically reduces the amount of transcytosed LTF, while favoring the accumulation of the protein in the nuclear area (Fig. 8b).

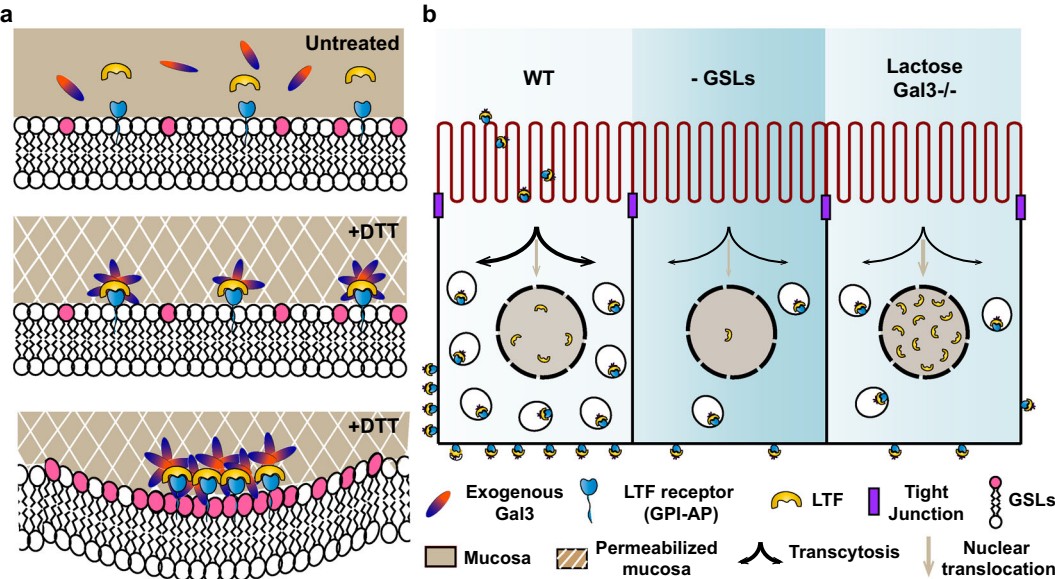

**Fig. 8 The GL-Lect hypothesis in mouse intestinal epithelium. a** Schematic representation of the GL-Lect hypothesis[9] transposed to the apical surface of enterocytes of the jejunum. In the absence of DTT-mediated permeabilization of the mucus, exogenous Gal3 and LTF do not gain access to the apical membrane surface and remain trapped in the mucus (top scheme). Upon DTT-mediated permeabilization of the mucus, monomeric Gal3 is recruited to apical membranes by binding to glycosylated cargo proteins on microvilli, such as LTF (middle scheme). Membrane-bound Gal3 oligomerizes and gains the functional capacity to interact with GSLs to drive membrane bending (bottom scheme). Gal3-mediated co-clustering of glycosylated cargo proteins and GSLs drives the formation of tubular endocytic pits from which CLICs are formed. **b** Schematic representation of the outcomes of the current study. Left: Transcytosis in control conditions. Middle: Under GSL depletion conditions, GL-Lect endocytosis of LTF is inhibited, leading to reduced transcytosis. Right: Under Gal3 KO conditions or in the presence of high lactose concentrations, GL-Lect endocytosis and transcytosis of LTF are inhibited, while LTF targeting to the nuclear region appears to be increased. See text for details.

Galectins are the key fabric of the GL-Lect mechanism[9]. Based on work with cells in culture, galectins have previously been implicated in apicobasal polarity, and basolateral-to-apical transcytosis[52–55]. The Gal3-dependent transcytotic apical-to-basolateral trafficking process in the intact jejunum of mice that we described in this study has not been reported before. In a previous study, the KO of Gal3 has been shown in enterocytes of the murine small intestine to reduce membrane polarization[56], which is consistent with the findings of our current study.

The other key fabric of the GL-Lect mechanism are GSLs. These have also been implicated in apicobasal polarity in the living organism[42]. Of note, direct evidence has been provided for the transcytotic trafficking of GSL-binding pathogenic lectins or modified versions of GSLs themselves[57–59]. Via some yet unidentified quality, GSLs appear to have the propensity to favor this process.

The discovery of GSL and Gal3-dependent transcytosis of a cellular cargo protein, LTF, as described here, argues in favor of a conceptual framework, the GL-Lect hypothesis, within which these previous findings can be analyzed from a fresh angle. This proposal may not be limited to the intestine. The GSL Gb3 has recently been shown to favor the reabsorption of filtered proteins from urine at the level of proximal tubules[60], suggesting that GL-Lect endocytosis and transcytosis may operate on scavenger proteins such as megalin/cubilin.

When galectin function was blocked, LTF transcytosis was inhibited, but its global uptake levels into enterocytes were unaltered. Under these conditions, more LTF was found in the nuclear area, suggesting that the protein was re-routed intracellularly. Evidence for DNA binding and nuclear localization of LTF has indeed been provided[61,62]. A likely interpretation of our findings is that GL-Lect endocytosis specifically couples to transcytotic trafficking, while an alternative form of endocytosis, possibly the clathrin pathway[63], would be linked to targeting of

the proteins into the nuclear area. In the absence of GSLs, Gal3 would still bind to LTF, which might prevent the latter from entering the alternative endocytic pathway. Collectively, our findings describe Gal3 as a new key player in transcytosis from apical to basolateral membranes of enterocytes.

The current study leads to the conclusion that GL-Lect endocytosis controls basolateral transcytotic processes in intestinal enterocytes. To what extent Gal3 function is seconded by other intestinal galectins, such as the abundant Gal4, the exact scope of cargoes whose transcytosis might be controlled by the GL-Lect mechanism, and the tissue contexts in which they operate even beyond the intestine are amongst the exciting questions that arise on the basis of the present work.

## Methods

**Materials**. Human Gal3-His, Cys-Gal3-His, and Gal3-TEV-His plasmids[7], 10-kDa and 40-kDa dextran (Chrondex), mouse LTF (Sino Biological), the clathrin-coated pit inhibitor ikarugamycin (SML0188, Sigma-Aldrich), the Gal3 inhibitor I3 (Hakon Leffler, Lund University, Sweden), and antibodies against: Gal3 (Fu-Tong Gal Liu, UC Davis, USA), ZO-1 (Abcam, Ref. ab59720, discontinued; Thermo Fisher, Ref. 61-7300), villin (Fatima El Marjou, Institut Curie, France), UEA-1 (Vector Laboratories), CD59 (H-7, sc 133170, Santa Cruz Biotechnology, Inc, USA), CD77 (IgM, Joelle Wiels, Institut Gustave Roussy, France), were obtained from the indicated people or commercial sources.

**Mouse lines and preparation of jejunum**. Wild-type, Gal3 KO[64], and conditional *Ugcg* KO mice[21] were of C57BL/6 background. The animals were generated and genotyped as described previously[21,64]. Approval of animal experiments and animal handling in a specific pathogen-free animal house facility strictly respected French regulations for animal care. "Comité d'éthique en expérimentation animale de l'institut Curie (CEEA-IC)". National registration number: #118.

In order to trigger the *Ugcg* gene deletion in *Ugcg* f/f/VilCre+/-ERT2 mice, these were injected intraperitoneally with the following solution. In an ultrasound bath, 1 mg of TAM was dissolved at 37 °C in 10 μL of pure ethanol, and 90 μL of sunflower seed oil (Sigma) was added with gentle mixing. Air bubbles were removed using an ultrasound bath. Three to 4-month-old male mice were used for all experiments. In order to identify the jejunum part of the intestine, the full intestine was removed and the first 8 cm starting from the pyloric sphincter were

cut and discarded since this region corresponds to the duodenum[65]. The remaining 16 cm were used and considered as the jejunum. The Swiss roll technique for histological studies of the mouse intestine was used to distinguish between intestinal regions[66].

**Genotyping by DNA extraction from mouse tails**. In total, 750 μL of the following solution were added to tails: 50 mM Tris-HCL, pH 8, 100 mM EDTA, 100 mM NaCl, 1% SDS, 20 μL proteinase K (10 μg/mL) (Sigma), followed by overnight incubation at 55 °C. 250 μL of saturated NaCl (>5 M) was added, followed by centrifugation for 20 min at 4 °C and 13,000 rpm in an Eppendorf centrifuge. In total, 750 μL of supernatant was supplemented with 2 μL RNAse A (10 μg/mL) (Thermo Fisher), and RNA was digested for 20 min at 37 °C. DNA was precipitated with 750 μL propanol-2, followed by centrifugation as above. After removal of the supernatant, 500 μL of ice-cold 80% ethanol was added and processed for centrifugation as above. The supernatant was removed, and Eppendorf tubes were inverted for 1 h to dry the DNA. 100 μL of distilled water was added, and DNA was resolved for 2 h at 37 °C.

**Recombinant Gal3 purification and labeling**. His-tagged Gal3 was purified as described[7]. For Alexa488 labeling, Gal3 (2 mg/mL) in PBS, 10 mM lactose was mixed with amine-reactive (NHS-ester) Alexa488 (Invitrogen) in a molar ratio of 1:4, and incubated for 1 h at 21 °C with agitation. The mixture was purified using PD-10 columns (GE Healthcare). Maleimide-activated horseradish peroxidase (HRP; Pierce) coupling to His-tagged Cys–Gal3 was performed for 12 h at 4 °C in PBS, 10 mM lactose at a molar ratio of 1:2. To separate HRP, Cys–Gal3–His, and HRP–Gal3–His, gel filtration chromatography was performed using a Superdex75 10 × 30 column. HRP–Gal3–His in 50% glycerol was snap-frozen and stored at −80 °C for further use.

**DTT treatment of intestine**. Mice were sacrificed by cervical dislocation, the jejunum was excised and washed with PBS until it became visually clean. The intestine was closed on one side with a clamp, injected with 10 mM DTT/PBS solution from the other side, and closed. Intestine segments were incubated for 15 min at room temperature, washed three times with DTT/PBS, and incubated again for 10 min with DTT/PBS. The intestine was washed three times with DTT/ PBS, and three times with PBS.

**Mucus fixation with Carnoy's solution and whole-mount immunofluorescence staining**. DTT-treated jejunum was fixed overnight in freshly prepared 60% ethanol, 30% chloroform, and 10% glacial acetic acid, cut with a scalpel in slices of ~1 mm, which were incubated for 1 h at room temperature in PBS, 1% Triton X-100, for 1 h at room temperature in PBS, 0.2% Triton X-100, 1% BSA, 3% FCS, overnight at room temperature with primary antibodies in PBS, 0.2% Triton X-100, washed three times for 1 h each with PBS, 0.2% Triton X-100, incubated overnight at room temperature with secondary antibodies in PBS, 0.2% Triton X-100, washed 3 times for 1 h each with PBS, 0.2% Triton X-100, mounted using an anti-fade mounting media (Thermo Fisher Scientific, USA), supplemented with DAPI, and stored at 4 °C.

**Ikarugamycin treatment**. DTT-treated jejunum segments were pre-incubated for 30 min at room temperature in PBS supplemented with 2 μM ikarugamycin (Ika). The compound was kept present in subsequent steps until the tissue was fixed.

**Binding and internalization assays**. DTT-treated jejunum segments were closed at one end with a clamp, and Gal3-Alexa488 (20 μg/mL), LTF–Cy3 (20 μg/mL), 40-kDa dextran (1 mg/mL), anti-CD59 or anti-CD77 antibodies (50 μg/mL), all diluted in PBS were injected from the other end, which was then also closed with a clamp. Segments were incubated for 30 min at 4 °C for the binding experiment, or continuously at 37 °C in PBS for the indicated periods of time. For pulse-chase experiments, segments were incubated for 30 min at 4 °C with the indicated ligands, washed once with PBS at 4 °C, and incubated at 37 °C for the indicated periods of time.

Jejunum segments were longitudinally opened, and excess of unbound ligands was washed out three times 10 min each at 4 °C with ice-cold PBS (except for CD59 and IgM, where only one wash of 5 min was performed). For Gal3 uptake experiments, ligands that remained cell surface-exposed were removed by three washes of 10 min each at 4 °C with ice-cold 200 mM lactose in iso-osmotic Hepes buffer. For binding experiments, the lactose step was omitted. Segments were directly washed, fixed, and processed for cryo-embedding, as described below.

**Tissue fixation and generation of frozen blocks and sections**. DTT-treated jejunum segments were incubated for 1 h at 4 °C in freshly prepared 4% PFA solution in PBS, then shifted to room temperature for 4 h, protected from the light. Tissue was washed three times 5 min with PBS, and then transferred to 30% sucrose (VWR chemicals, D(+)-saccharose) solution in PBS at 4 °C. After overnight incubation, tissue was transferred to 100% optimal cutting temperature compound solution in plastic molds (Sakura). Molds with tissue were located onto pieces of dry ice. When all of the optimal cutting temperature compounds turned

white, the tissue (still in the plastic mold) was placed in the −80 °C freezer and stored until use. Frozen sections were prepared on a cryostat (Leica CM 1950). To mount tissue blocks on the specimen holder, some optimal cutting temperature compound was added on top of the metal holder inside the cryotome. As it started to freeze, it was popped out of the frozen block containing the tissue and positioned flatly on the optimal cutting temperature compound. Sections of 25 μm were mounted on slides (Superfrost Plus, Thermo Scientific).

**Immunohistochemistry**. Slides were thawed for 5 min at room temperature, rehydrated in PBS for 5 min, incubated for 10 min in 50 mM NH4Cl/PBS, permeabilized for 15 min with PBS, 0.2% Triton X-100, incubated for 2 h at room temperature in blocking buffer (PBS, 2% BSA, 0.2% Triton X-100), and overnight at 4 °C with primary antibodies diluted in blocking buffer. After 3 washes in PBS for 5 min each, slides were incubated 60 min at room temperature with secondary antibodies diluted in PBS, 2% BSA, 0.2% Triton X-100. After three washes in PBS for 5 min each, sides were mounted with an anti-fade mounting media (Thermo Fisher Scientific, USA) containing DAPI (Thermo Fisher Scientific, USA), and visualized on a Nikon A1R confocal microscope with a ×60 oil immersion objective. For quantification, confocal images were background subtracted, cellular regions were circled, and total fluorescence intensity was measured using the ImageJ program.

**Electron microscopy**. DTT-treated jejunum segments were incubated at 37 °C for the indicated times with 10 mg/mL HRP in PBS, PBS alone, or 40 μg/mL HRP-Gal3 in PBS, or at 4 °C for cell surface binding. Cell surface-exposed markers were removed by three washes for 10 min each at 4 °C with 200 mM lactose in iso-osmotic HEPES buffer (step omitted for binding experiments). The intestine was opened in transverse orientation, cut in 5 × 8-mm pieces, incubated for 15 min at 4 °C in freshly prepared 0.7 mg/mL 3,3'-diaminobenzidine (DAB) solution (Sigma), incubated for 1 h at 4 °C with DAB + 30% H2O2, washed three times 10 min each with PBS, fixed for 2–3 days at 4 °C with 2.5% glutaraldehyde (EMS) in 0.1 M Na-cacodylate (pH 7.2), washed three times for 5 min each at room temperature with 0.1 M Na-cacodylate, post-fixed for 90 min at room temperature with 1% OsO4 (EMS) in 0.1 M Na-cacodylate, washed three times for 10 min each with 0.1 M Na-cacodylate, followed by a final washing for 5 min with water. Tissue dehydration was performed at room temperature according to the following incubation protocol: 50% ethanol 10 min, 70% ethanol 10 min, 90% ethanol two times 15 min each, 100% ethanol three times 20 min each, followed by infiltration of Epon resin LX112 at room temperature (LX112/EtOH 100% 1v/1v for 30 min, LX112/EtOH 100% 2v/1v for 1 h, LX112 overnight). The polymerization of tissue in LX112 was done in embedding mold (tissue bloc 2 × 3 mm) for 2–3 days at 60 °C. Samples were sectioned in an ultramicrotome (Reichert Leica UCT), and sections of 65 nm thickness were deposited on formvar/carbon-coated grid (form/carbon square 100 mesh – Cu: FCF100-CU-SB EMS). The contrast was enhanced by incubation for 10 min with 4% uranyl acetate in water. Sections were imaged using a Tecnai Spirit electron microscope (FEI, Eindhoven, The Netherlands) equipped with a 4K CCD camera (EMSIS GmbH, Münster, Germany).

**Identification of galectins content in the jejunum by MS**. Enterocytes were scraped off the lamina propria with a glass slide (Superfrost, Thermo Scientific). 1 mL lysis buffer (PBS, 2% Triton X-100, 0.5% NP40) was added to the cells, which were passed ten times through a 23G needle. The lysate was cleared by 16,000×g centrifugation for 10 min at 4 °C.

Lactose agarose beads (L7634-5ml, Sigma) were washed two times in lysis buffer and incubated with lysates by end-over-end rotation for 1 h at 4 °C. Beads were then washed in lysis buffer, washed twice with 100 μL of 25 mM NH4HCO3. Bound galectins were digested on beads with 0.2 μg of trypsin/LysC (Promega) for 1 h in 50 μL of 25 mM NH4HCO3 buffer. Samples were desalted and analyzed by nanoLC-MS/MS using an UltiMate 3000 RSLCnano system (Thermo Scientific) coupled to an Orbitrap Fusion (Q-OT-qIT, Thermo Fisher Scientific) mass spectrometer[67].

**Identification of Gal3 interacting partners by MS**. DTT-treated jejunum segments were incubated for 30 min at 4 °C in PBS alone, or PBS containing 20 μg/mL of Gal3–His. After two washes with PBS, the intestine was longitudinally opened, enterocytes of the jejunum were scraped off with a glass slide (Thermo Scientific, USA) and lysed with PBS, 2% Triton X-100, 0.5% NP40 in the presence of protease inhibitor cocktail. After centrifugation at 16,000×g for 30 min at 4 °C, cleared lysates were incubated overnight at 4 °C with cobalt beads, washed with lysis buffer (without protease inhibitors), and sent to MS for further analysis.

Gal3 interacting partners were eluted with two buffers: (1) loading buffer for SDS PAGE (LSB 1×): 2% SDS 1×; (2) HEPES 50 mM, NaCl 50 mM, lactose 200 mM, pH 7.5 and LSB 1×, and separated by SDS-PAGE (30 min, 80 V). After staining with colloidal blue, complete lanes of proteins were excised for each replicate (n = 3) and processed. Excised gel slices were washed, and proteins were reduced with 10 mM DTT prior to alkylation with 55 mM iodoacetamide. After washing and shrinking of the gel pieces with 100% acetonitrile, in-gel digestion was performed overnight at 30 °C using trypsin/LysC (0.1 μg) in 25 mM ammonium bicarbonate. Peptides were then extracted using 60/35/5 MeCN/H2O/HCOOH,

vacuum concentrated to dryness, and samples were loaded onto homemade C18 StageTips for desalting. Peptides were eluted from beads by incubation with 40/60 MeCN/H$_2$O + 0.1% formic acid. The peptides were dried in a Speedvac and reconstituted in 10 μL 2/98 MeCN/H$_2$O + 0.3% trifluoroacetic acid (TFA) prior to liquid chromatography-tandem MS (LC-MS/MS) analysis. Samples were chromatographically separated using an RSLCnano system (Ultimate 3000, Thermo Scientific) coupled online to a Q Exactive HF-X with a Nanospray Flex ion source (Thermo Scientific). Peptides were trapped onto a C18-reversed-phase precolumn (75-μm inner diameter × 2 cm; nanoViper Acclaim PepMap™ 100, Thermo Scientific), with buffer A (2/98 MeCN/H$_2$O + 0.1% formic acid) at a flow rate of 2.5 μL/min over 4 min. The separation was performed on a 50 cm × 75-μm C18 column (nanoViper Acclaim PepMap™ RSLC, 2 μm, 100 Å, Thermo Scientific) regulated to a temperature of 50 °C with a linear gradient of 2% to 30% buffer B (100% MeCN and 0.1% formic acid) at a flow rate of 300 nL/min over 91 min. Full MS scans were performed in an ultrahigh-field Orbitrap mass analyzer in ranges m/z 375–1500 with a resolution of 120,000 at m/z 200. The maximum injection time (MIT) was 50 ms, and the automatic gain control (AGC) was set to 3 × 10$^6$. The top 20 intense ions were subjected to Orbitrap for further fragmentation via high energy collision dissociation (HCD) activation and a resolution of 15,000 with the intensity threshold kept at 1.3 × 10$^5$. We selected ions with charge state from 2+ to 6+ for screening. Normalized collision energy (NCE) was set at 27. For each scan, the AGC was set at 1 × 10$^5$, the MIT was 60 ms, and the dynamic exclusion of 40 s.

For identification, the data were searched against the UniProt *Mus musculus* canonical (house Mouse downloaded on 22/08/2017) database using Sequest-HT through proteome discoverer (version 2.0). Enzyme specificity was set to trypsin, and a maximum of two missed cleavage sites was allowed. Oxidized methionine, N-terminal acetylation, and carbamidomethyl cysteine were set as variable modifications. Maximum allowed mass deviation was set to 10 ppm for monoisotopic precursor ions and 0.02 Da for MS/MS peaks. The resulting files were further processed using myProMS[68] v3.9. FDR calculation used Percolator and was set to 1% at the peptide level for the whole study. The label-free quantification was performed by peptide Extracted Ion Chromatograms (XICs) computed with MassChroQ version 2 (ref. [69]). For protein quantification, XICs from proteotypic peptides shared between compared conditions (TopN matching) with missed cleavages and carbamidomethyl modifications were used. Median and scale normalization was applied on the total signal to correct the XICs for each biological replicate. To estimate the significance of the change in protein abundance, a linear model (adjusted on peptides and biological replicates) was performed, and p-values were adjusted with a Benjamini–Hochberg FDR procedure with a control threshold set to 0.05. To focus from this complex list on partners that were only present in Gal3 replicates, we selected in the qualitative comparison proteins with at least one peptide in each Gal3 replicate, zero peptide in the PBS control, and only based on proteotypic peptides, which allowed to capture specific proteins.

**In vitro interaction studies between Gal3 and LTF**. Purified mouse LTF-His and human Gal3-Cy3 without a His-tag (4 μg/mL each) were incubated in PBS supplemented with 0.1% Tween-20 (PBS-T) buffer under agitation for 15 min at 18 °C, in the presence or absence of the Gal3 inhibitor I3. Cobalt beads (#89965, Thermo Scientific) were added for 30 min incubation under rotation at 4 °C, collected by centrifugation at 700×g, washed three times with PBS-T buffer, and eluted by incubation for 30 min at room temperature under agitation with 200 mM lactose solution, or by boiling in sample buffer for 10 min at 95 °C. Eluted proteins were analyzed on Stain-Free™ SDS-PAGE gels (Bio-Rad).

**TLC and LC-MS/MS analysis of lipids**. Freshly dissected tissue was lyophilized in 2 mL Eppendorf tubes, and dry weight was determined. After the addition of a steel bullet and 500 μL chloroform:methanol:water (10:10:1), tissue was homogenized with a TissueLyser II (Qiagen) at 25 Hz for two times 2 min. After centrifugation in a tabletop instrument with 13,000 rpm at room temperature, the supernatant was collected in a new glass tube. The pellet was suspended in 500 μL chloroform: methanol:water (10:10:1), incubated for 5 min at 37 °C in an ultrasound bath and centrifuged again. The supernatant was collected, the residual pellet was suspended with chloroform:methanol:water (30:60:8), and incubated for 5 min at 37 °C in an ultrasound bath. After centrifugation, the supernatant was collected and dried with a gentle nitrogen stream at 37 °C. Samples were dissolved at 20 μL/mg dry weight in chloroform:methanol:water (10:10:1). For TLC, 15 μL corresponding to 0.75 mg tissue dry weight was loaded on each lane together with GSL standards for GlcCer, LacCer, GA2, and GA1. TLC was predeveloped with chloroform:acetone (1:1), dried thoroughly, and then developed with the solvent system chloroform: methanol:0.2% aqueous CaCl$_2$ (60:35:8). After drying, TLC was sprayed with orcinol reagent and incubated for ~10 min at 120 °C to stain glycolipids. For LC-MS/MS analysis, an aliquot corresponding to 2 mg tissue dry weight was mixed with an internal GlcCer standard (GlcCer(d18:1/14:0), GlcCer(d18:1/19:0), GlcCer (d18:1/25:0), and GlcCer(d18:1/31:0), each at a concentration of 50 pmol/sample). LC-MS/MS measurements were performed on an Acquity I-class UPLC and a Xevo TQ-S "triple quadrupole" instrument, both from Waters. Using a CSH C18 column (2.1 × 100 mm, 1.7 μm; Waters), lipids were measured in reversed-phase-LC mode with a gradient between 70% solvent A (50% methanol) and 99% solvent B

(1% methanol, 99% isopropanol), both containing 0.1% formic acid and 10 mM of ammonium formate as additives. Lipids were detected by multiple reaction monitoring (MRM). GlcCers were detected with the transition to their sphingoid base fragment, and GA1 was detected with the loss of the terminal disaccharide ion. Both, GlcCer, and GA1 with ceramide anchors containing a C18-sphingosine or C18 phytosphingosine in combination with either non-hydroxy or α-hydroxy fatty acids (C16:0, C18:0, C20:0, C22:0, C23:0, C24:1, C24:0) were monitored.

**Statistics and data reproducibility**. All experiments were done with at least three biological replicates (except when indicated). All tissue quantifications were performed with at least 34 cells per condition. For each experiment, we show representative data sets. Statistical analysis was performed either by unpaired Student's *t* test, or with ordinary one-way ANOVA (for multiple comparisons).

**Reporting summary**. Further information on research design is available in the Nature Research Reporting Summary linked to this article.

## Data availability
The authors declare that all data supporting the findings of this study are available within the paper and its supplementary information files. All MS proteomics data have been deposited to the ProteomeXchange Consortium via the PRIDE partner repository[70] with the dataset identifier PXD016499 (username: reviewer18055@ebi.ac.uk, password: ARHPfQ1I). All source data underlying the graphs presented in the main and Supplementary Figs. are available via the following Figshare repository link[71]: https://ndownloader.figshare.com/files/25713287 (login: massiullah.shafaq-zadah@curie.fr, password: UMR3666-InstitutCurie). All other data generated and/or analyzed during the current study are available from the corresponding author upon reasonable request.

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

## Acknowledgements

We would like to thank the following people for help in experiments, providing materials, and/or expertise: Marina Glukhova, Anna Zagryazhskaya-Masson, Danijela Vignjevic, Hakon Leffler, Florent Dingli, Guillaume Arras, Caspar Caspersen, Fu-Tong Liu, Denis Krndija, Fatima El Marjou, *Institute Curie*, and *Institut Jacques Monod* mouse facility members. We acknowledge support by grants from the *Agence Nationale pour la Recherche* (ANR-14-CE14-0002-02, ANR-16-CE23-0005-02, ANR-19-CE13-0001-01), Human Frontier Science Program (RGP0029-2014), European Research Council (advanced grant 340485), the Swedish Research Council (K2015-99X-22877-01-6). The Johannes team is a member of Labex Cell(n)Scale (11-LBX-0038) and Idex Paris Sciences et Lettres (ANR10-IDEX-0001-02 PSL). We would also

like to acknowledge the Cell and Tissue Imaging (PICT-IBiSA) and Nikon Imaging Centre, *Institut Curie*, member of the French National Research Infrastructure France-BioImaging (ANR10-INBS-04).

## Author contributions

M.S.Z. and L.J. designed the study. M.S.Z. and C.W. directed the experimental work. A.I. did all mouse experiments. V.C. all electron microscopy experiments. R.S., B.L., and D.L.: all mass spectrometry experiments. E.D. monitored reproducibility. R.J. and H.J.G. provided expertise on *ugcg* KO mouse model, K.P. on cytochemistry and animal handling, C.L. on endocytosis, and F.P. on galectins.

## Competing interests

The authors declare no competing interests.
