## [Peer Review File · Communications Biology]

Reviewers' comments:

Reviewer #1 (Remarks to the Author):

This paper is concerned with a new mode of endocytosis that involves glycosphingolipids and lectins, GL-Lect endocytosis that the Johannes lab has been studying. This endocytic pathway is used by pathogenic lectins such as Shiga toxin but also by endogenous lectins such as galectins. Until now this work has relied on cell culture and in vitro models. The main objective here was to analyze the galectin-3 and glycosphingolipid dependency of the GL-Lect endocytosis route in mouse enterocytes in the jejunum of the small intestine of C57BL6 mice. Using electron microscopy, the endocytosis of horse radish peroxidase-coupled galectin-3 could be followed across the enterocytes and the intermediate stations identified. To study whether galectin-3 indeed was dependent on glycosphingolipids for its transcytosis, the enzyme regulating glucosylceramide-based glycosphingolipids was genetically eliminated from the intestine of the mice. This led to decreasing levels of glucosylsphingolipids that function as receptors for galectin-3. As a consequence transcytosis of galectin-3 was decreased. They also identified a protein interacting with galectin-3, lactotransferrin and showed that lactotransferrin transcytosed together with galectin-3 to the basolateral side in a glycosphingolipid-dependent manner. Thus this protein found in mother's milk could be taken up in the circulation of the baby and be part of their infection defense.

Overall, the paper is interesting and the experiments are mostly well done. I have only one comment and it can only be answered by additional experiments. The authors claim to have demonstrated transcytosis of galectin-3 and of lactotransferrin. However, no experiment was done to show that the glycolipid-lectin complex indeed arrived at the basolateral surface. All one can say from the IF and EM experiments is that the complex reached that basolateral region from the apical surface. Transcytosis means that the ligands should surface on the basolateral side. The authors need to do double-label experiments; in which both the basolateral membrane and galectin-3 or lactotransferrin are labeled. If the labels were to overlap, they would prove their point.

Reviewer #2 (Remarks to the Author):

Ivashenka, Johannes, Shafaq-Zadah and colleagues are reporting that galectin-3 (Gal3) was binding to lactotransferrin (LTF) to drive its transcytosis (i.e. transport from the apical to basolateral membrane) in enterocytes. Such trafficking was Gal3- and Glycosphingolipid (GSL)-dependent, and Gal3 was found in clathrin-independent carriers (CLICs), consistent with the glycolipid-lectin (GL-Lect) hypothesis proposed by the Johannes lab. This is new and will interest the scientific community interested in membrane trafficking, iron metabolism and glycosphingolipids.

The authors used multiple assays to provide clear evidence supporting their conclusions. Overall, the study was well designed, executed at a high standard and analysed carefully. Although more physiological, ex vivo experiments are arguably less precise than in vitro assays in cultured cells. Thus, few controls are missing and should be added to strengthen the conclusions and rule out alternative explanations.

- 1) Figure 4a-b: the authors should show a lipid that does not decrease upon Cre activation to rule out general and unspecific effects.
- 2) Figure 4c-e: a marker that is not affected by the Cre activation should be added to one of the panel to rule out overall loss of cell integrity
- 3) Figure 5: same concern: a marker (transferrin?) which uptake is not affected by the Cre activation should be added to rule out overall loss of cell integrity. The total loss of Dextran uptake is concerning because one would not expect all endocytosis to be inhibited upon removal of GSLs.

There is a risk that the epithelium permeability was lost and thus the dextran diffused out. Perhaps the authors could try higher doses? (31 ug/mL is low compared to the 1-2mg/mL used in other publications).

4) The 4°C pre-incubation will likely blunt other CIE pathways (FEME and macropinocytosis in particular). Adding one experiment omitting the cold step and/or inhibiting Dynamin would add to the paper to confirm that Gal3-LTF enter cells through the CLIC/GEEC pathway.

5) Mechanistically, it is not clear whether Gal3-LTF are unique in doing transcytosis or whether all cargoes entering through the GL-Lect mechanism in enterocytes are directed to the basolateral side. Looking for the fate of a known cargo entering through the pathway (CD44, CD59 etc..) would be informative.

Minor comments:

- Model (Figure 8): the LTF receptor is unlikely to be a transmembrane protein as Intelectin-1 is a GPI-anchored protein and GAPDH is soluble and does not have TM regions

- Figure 3a and b would benefit from quantitation.

Reviewer #3 (Remarks to the Author):

In this manuscript, Ivanshenka and colleagues show that Lactotransferrin (LTF) undergoes a glycolipid- and lectin-mediated transcytosis in mouse enterocytes. This study follows-up the discovery, from this group, that selected molecules at the plasma membrane are internalized via clathrin-independent endocytosis in a process that requires sphingolipids and members of the Galectin family of lectins (also known as the Glyco-Lectin hypothesis). Here, the authors use an ex-vivo model system based on the mouse jejunum excised from either conditional or knockout mouse models. They demonstrate that in the enterocytes LTF undergoes internalization from the apical plasma membrane and transcytosis to the basolateral membrane through its interaction and binding to Galectin3 and glycosphingolipids.

The work described in this manuscript supports the conclusions from the authors and it is of significant importance since it extends the conclusions from previous work in cell cultures to a more physiological system.

This reviewer has only a few suggestions/comments to strengthen the presentation of the overall story.

1) Figure 3 – panels a. The authors should show a micrograph at time 0 (i.e. incubation with Gal3-488 at 4°C and wash-out with lactose before switching to 37°C deg). This would further confirm that the signal at 5 min is indeed internalized material and not background.

2) Figure 3 – panels b. I recommend replacing the EM micrographs with others of better quality. It is clear in the selected insets that the vacuoles in the controls ("No HRP") do not show DAB precipitation. However, it is not so in the remaining field of view where it seems that Dab precipitation occurs also in the controls.

3) In Fig. 4, the quantification of the GSL level should come from 3 replicates rather than 2, to achieve statistical significance.

4) The figure legends should be more consistent in describing the number of experiments. The authors should use the same format since it is not always clear how many independent replicates and how many cells per replicate have been analyzed.

5) Please, re-check the text for typos and grammar.

We are grateful for the constructive comments by the reviewers, which have helped us to further improve our submitted manuscript. We have performed many new experiments, which has taken longer than initially planned due to the lockdown of our animal facility during the COVID-19 confinement period in France, and the need to grow up the animal lines afterwards. Our responses to the initial reviewer comments are shown below in red and italics.

Reviewers' comments:

Reviewer #1 (Remarks to the Author):

This paper is concerned with a new mode of endocytosis that involves glycosphingolipids and lectins, GL-Lect endocytosis that the Johannes lab has been studying. This endocytic pathway is used by pathogenic lectins such as Shiga toxin but also by endogenous lectins such as galectins. Until now this work has relied on cell culture and in vitro models. The main objective here was to analyze the galectin-3 and glycosphingolipid dependency of the GL-Lect endocytosis route in mouse enterocytes in the jejunum of the small intestine of C57BL6 mice. Using electron microscopy, the endocytosis of horse radish peroxidase-coupled galectin-3 could be followed across the enterocytes and the intermediate stations identified. To study whether galectin-3 indeed was dependent on glycosphingolipids for its transcytosis, the enzyme regulating glucosylceramide-based glycosphingolipids was genetically eliminated from the intestine of the mice. This led to decreasing levels of glucosylsphingolipids that function as receptors for galectin-3. As a consequence, transcytosis of galectin-3 was decreased. They also identified a protein interacting with galectin-3, lactotransferrin and showed that lactotransferrin transcytosed together with galectin-3 to the basolateral side in a glycosphingolipid-dependent manner. Thus, this protein found in mother's milk could be taken up in the circulation of the baby and be part of their infection defense.

Overall, the paper is interesting and the experiments are mostly well done. I have only one comment and it can only be answered by additional experiments. The authors claim to have demonstrated transcytosis of galectin-3 and of lactotransferrin. However, no experiment was done to show that the glycolipid-lectin complex indeed arrived at the basolateral surface. All one can say from the IF and EM experiments is that the complex reached that basolateral region from the apical surface. Transcytosis means that the ligands should surface on the basolateral side. The authors need to do double-label experiments; in which both the basolateral membrane and galectin-3 or lactotransferrin are labeled. If the labels were to overlap, they would prove their point.

*To answer reviewer's remark on the basolateral localization of Gal3 or LTF after their uptake from the apical side, we have performed the following new experiment: Gal3-488 was continuously incubated for 30 min at 37 °C with the jejunum preparation. After having removed the surface exposed pool of Gal3 with lactose and tissue fixation, we have immunostained the sample for the adherens junction marker E-cadherin. **We confirmed that E-cadherin indeed labeled the basolateral domain in our preparation, and found that cadherin labeling strongly overlapped with internalized Gal3.** This new set of data is now documented in **Fig. 3b**.*

Reviewer #2 (Remarks to the Author):

Ivashenka, Johannes, Shafaq-Zadah and colleagues are reporting that galectin-3 (Gal3) was binding to lactotransferrin (LTF) to drive its transcytosis (i.e. transport from the apical to basolateral membrane) in enterocytes. Such trafficking was Gal3- and Glycosphingolipid (GSL)-dependent, and Gal3 was found in clathrin-independent carriers (CLICs), consistent with the glycolipid-lectin (GL-Lect) hypothesis proposed by the Johannes lab. This is new and will interest the scientific community interested in membrane trafficking, iron metabolism and glycosphingolipids.

The authors used multiple assays to provide clear evidence supporting their conclusions. Overall, the study was well designed, executed at a high standard and analyzed carefully. Although more physiological, ex vivo experiments are arguably less precise than in vitro assays in cultured cells. Thus, few controls are missing and should be added to strengthen the conclusions and rule out alternative explanations.

1) Figure 4a-b: the authors should show a lipid that does not decrease upon Cre activation to rule out general and unspecific effects.

*To rule out a general perturbation of lipid metabolism upon depletion of the *ucgc* gene, we have quantified the ceramide-containing but GSL-unrelated sphingomyelin (SM) by LC-MS/MS. **Importantly, the level of SM was not significantly altered in *Ugcgflox/Cre⁺* mice for all time points after tamoxifen injection that we have tested in our study. This new set of data is reported in Fig. 4b.***

2) Figure 4c-e: a marker that is not affected by the Cre activation should be added to one of the panel to rule out overall loss of cell integrity

*There might be a misunderstanding here that we would like to clarify. These figures indeed are built exactly to rule out a loss of cell integrity. For this, we document that even in *Ugcgflox/Cre⁺* mice, the apically located villin as well as the sub-apically located tight junction marker ZO-1 are localized normally (i.e. indistinguishably from control mice) at days 1 and 2 after tamoxifen injection, when GSL levels are already strongly reduced. We do not consider day 3 after tamoxifen injection, at which indeed a massive loss of cell integrity is observed. Day 3 results are only shown for completeness. We do not draw any conclusions from them as to the endocytosis and transcytosis processes that are the subject of our submission.*

*Tissue integrity at days 1 and 2 after tamoxifen injection was also analyzed by incubating the jejunum preparation at 4 °C with Gal3-488. At this temperature, endocytosis is inhibited, and an appearance of Gal3 at the lateral domain could only come from a loss of tight junction integrity. However, this was not observed: No paracellular leakage of Gal3 was detected for up to 2 days after tamoxifen injection in the *Ugcgflox/Cre⁺* genetic background (Fig. 5a). At day 3, a dramatic loss of Gal3 binding was observed, which again reflects on the deterioration of the tissue at this time point. As mentioned above, this time point was not taken into consideration for the conclusions of our study.*

In summary, several conditions were used to establish tissue integrity at days 1 and 2 after tamoxifen injection and subsequent GSL depletion. The day 3 time point was not taken into account for the conclusions of our study. This aspect was better clarified in the revised version of the manuscript.

3) Figure 5: same concern: a marker (transferrin?) which uptake is not affected by the Cre activation should be added to rule out overall loss of cell integrity. The total loss of Dextran uptake is concerning because one would not expect all endocytosis to be inhibited upon removal of GSLs. There is a risk that the epithelium permeability was lost and thus the dextran diffused out. Perhaps the authors could try higher doses? (31 ug/mL is low compared to the 1-2mg/mL used in other publications).

To avoid any confusion: Dextran uptake is not inhibited at days 1 and 2 after Cre activation and GSL depletion, which are the only time points that we consider for the conclusions of our study (see above).

*Transferrin could not be used as a marker since its receptor is located at the basolateral pole and therefore, it is not accessible for endocytic uptake from the lumen of the intestine. As an alternative, we have chosen to work on another established clathrin-dependent cargo protein, immunoglobulin M (IgM). No alteration of IgM internalization was observed at days 1 and 2 post-tamoxifen injection (**Fig. S2a**), which documents that, as expected, the clathrin machinery is still functional in the absence of GSLs. In contrast, IgM uptake was strongly reduced when the clathrin machinery was inhibited with a small molecule compound, ikarugamycin, while the internalization of Gal3, LTF, and CD59 (see below), was not or only little affected. This entirely new data is reported in **Fig. S1**.*

*To address the point on dextran concentration, we repeated our experiments at a dextran concentration of 1 mg/ml. Also, at this concentration, dextran entered GSL-depleted cells (days 1 and 2 after tamoxifen injection in *Ugcglox/Cre⁺* mice) as efficiently as in control conditions (days 1 and 2 after tamoxifen injection in *Ugcglox/Cre⁻* mice), as we had reported before at the lower concentrations of dextran. In both cases (i.e. GSL-depleted and non-depleted cells), dextran accumulated into subapical vacuolar-shaped endocytic structures. We have replaced the data of previous **Fig. 5c** with the newly acquired ones at 1 mg/ml of dextran.*

4) The 4°C pre-incubation will likely blunt other CIE pathways (FEME and macropinocytosis in particular). Adding one experiment omitting the cold step and/or inhibiting Dynamin would add to the paper to confirm that Gal3-LTF enter cells through the CLIC/GEEC pathway.

There might be a misunderstanding here that we would like to clarify. The 4 °C step was only used in the following conditions:

*- Pulse-chase experiment as reported in **Fig. 3a**, in order to evaluate the gradual intracellular progression of apically internalized Gal3.*

- Binding experiments, notably to make sure that our experimental procedures (Gal3 or Ugcg KO, DTT or ikarugamycin treatments...) do not perturb the plasma membrane levels of expression of our cargoes of interest, and that the tissue is still intact (no leakage).

For all other functional assays and especially the endocytosis and transcytosis experiments, we rigorously performed continuous incubations at 37 °C, without any passage at 4 °C.

*To further confirm that Gal3 and LTF enter the cells by clathrin-independent endocytosis, we have used ikarugamycin to inhibit clathrin-coated pit formation. Interestingly, the cellular entry of both protein cargoes was only mildly (Gal3) or not (LTF) affected, confirming a major dependency on non-clathrin endocytosis for uptake. These entirely new data are shown in **Fig. S1a and S1d**.*

5) Mechanistically, it is not clear whether Gal3-LTF are unique in doing transcytosis or whether all cargoes entering through the GL-Lect mechanism in enterocytes are directed to the basolateral side. Looking for the fate of a known cargo entering through the pathway (CD44, CD59 etc..) would be informative.

*As suggested by the reviewer, we have tested the uptake of CD59 which is apically localized in mouse enterocytes. Using an antibody uptake assay, we found that internalized CD59 did not exhibit a transcytotic pattern, but rather was localized to subapical punctate-like structures, as reported in new **Fig. S1c**. As expected, the uptake of CD59 was found to be clathrin-independent, as it was not inhibited by ikarugamycin (**Fig. S1c**). It was not dependent on GSLs either (**Fig. S2b**), likely because the GPI-anchored protein CD59 is a glycolipid itself.*

Minor comments:

- Model (Figure 8): the LTF receptor is unlikely to be a transmembrane protein as Intellectin-1 is a GPI-anchored protein and GAPDH is soluble and does not have TM regions

We thank reviewer for these valuable indications. The model has been modified accordingly.

- Figure 3a and b would benefit from quantitation.

We have now provided these quantifications.

Reviewer #3 (Remarks to the Author):

In this manuscript, Ivanshenka and colleagues show that Lactotransferrin (LTF) undergoes a glycolipid- and lectin-mediated transcytosis in mouse enterocytes. This study follows-up the discovery, from this group, that selected molecules at the plasma membrane are internalized via clathrin-independent endocytosis in a process that requires sphingolipids and members of the Galectin family of lectins (also known as the Glyco-Lectin hypothesis). Here, the authors use an ex-vivo model system based on the mouse jejunum excised from either conditional or knockout mouse models. They demonstrate that in the enterocytes LTF undergoes internalization from the apical plasma membrane and transcytosis to the basolateral membrane through its interaction and binding to Galectin3 and glycosphingolipids.

The work described in this manuscript supports the conclusions from the authors and it is of significant importance since it extends the conclusions from previous work in cell cultures to a more physiological system.

This reviewer has only a few suggestions/comments to strengthen the presentation of the overall story.

1) Figure 3 – panels a. The authors should show a micrograph at time 0 (i.e. incubation with Gal3-488 at 4°C and wash-out with lactose before switching to 37°C deg). This would further confirm that the signal at 5 min is indeed internalized material and not background.

As requested by the reviewer, we have added the zero time point, and also a quantification of these data to the revised version of the figure (Fig. 3a).

2) Figure 3 – panels b. I recommend replacing the EM micrographs with others of better quality. It is clear in the selected insets that the vacuoles in the controls (“No HRP”) do not show DAB precipitation. However, it is not so in the remaining field of view where it seems that Dab precipitation occurs also in the controls.

We replaced several of the EM images to further improve the quality of the figure.

As for DAB, we indeed also observed precipitates in the control (“no HRP”) condition. However, these precipitates were found mainly in the cytosol (no signal is visible within the vacuolar structures), and they were large in size (>125 nm). In contrast, in the Gal3-HRP situation, smaller precipitates (approx. 50 nm) were seen in addition to the larger ones, and the former were localized to vacuolar or tubular structures. This quantification is now inserted in the revised figure (Fig. 3c). We interpret this data as to show that the small vacuole/tubule-localized precipitates that are only found in the Gal3-HRP condition are the endocytic ligand-specific signals.

3) In Fig. 4, the quantification of the GSL level should come from 3 replicates rather than 2, to achieve statistical significance.

We do agree with the reviewer's suggestion and we have now provided 3 biological replicates and achieved statistical significance. We have therefore replaced the previous graph by an updated version, in which we also quantified the expression of another lipid, sphingomyelin, which was not affected by Ugcg deletion, thereby demonstrating the specificity of the phenotype (Fig. 4b).

4) The figure legends should be more consistent in describing the number of experiments. The authors should use the same format since it is not always clear how many independent replicates and how many cells per replicate have been analyzed.

The figure legends have been revised as requested by the reviewer to describe the number of experiments and to indicate clearly how many independent replicates have been analyzed.

5) Please, re-check the text for typos and grammar.

The text was checked again for typos and grammar by 2 native English speakers.

REVIEWERS' COMMENTS:

Reviewer #1 (Remarks to the Author):

The experiment done answered my question. From my side the paper can now be recommended for publication.

Reviewer #2 (Remarks to the Author):

I thank the authors for addressing every single of my comments and for their detailed answers and clarifications.

They significantly improved their manuscript by adding many new data and updating the text. I have no further concerns and I congratulate all the authors for their important contribution.

Reviewer #3 (Remarks to the Author):

The authors have fully addressed my concerns and the study is significantly improved.